


# A missing link in the carbon cycle: phytoplankton light absorption

Rémy Asselot[1,*], Frank Lunkeit[2], Philip Holden[3], and Inga Hense[1]

[1]Institute for Marine Ecosystem and Fishery Science, Center for Earth System Research and Sustainability, University of Hamburg, Hamburg, Germany
[2]Meteorological Institute, Center for Earth System Research and Sustainability, University of Hamburg, Hamburg, Germany
[3]Environment, Earth and Ecosystems, The Open University, Walton Hall, Milton Keynes, MK7 6AA, UK
[*]Now at Ifremer, University of Brest, CNRS, IRD, Laboratoire d'Océanographie Physique et Spatiale (LOPS), UMR6523, Centre de Bretagne, 29280 Plouzané, France

**Correspondence:** Rémy Asselot (remy.asselot@ifremer.fr)

**Abstract.** Marine biota and biogeophysical mechanisms, such as phytoplankton light absorption, have attracted increasing attention in recent climate studies. Under global warming, the impact of phytoplankton on the climate system is expected to change. Previous studies analyzed the impact of phytoplankton light absorption under prescribed future atmospheric $CO_2$ concentrations. However, the role of this biogeophysical mechanism under freely-evolving atmospheric $CO_2$ concentration and future $CO_2$ emissions remain unknown. To shed light on this research gap, we perform simulations with the EcoGEnIE Earth system model and prescribe $CO_2$ emissions following the four Representative Concentration Pathways (RCP) scenarios. Under all the RCP scenario, our results indicate that phytopankton light absorption increases the surface chlorophyll biomass, the sea surface temperature, the atmospheric $CO_2$ concentrations and the atmospheric temperature. Under the RCP2.6, RCP4.5 and RCP6.0 scenarios, the magnitude of changes due to phytoplankton light absorption are similar. However, under the RCP8.5 scenario, the changes in the climate system are less pronounced due to the temperature limitation of phytoplankton growth, highlighting the reduced effect of phytoplankton light absorption under strong warming. Additionally, this work evidences the major role of phytoplankton light absorption on the climate system, suggesting a highly uncertain feedbacks on the carbon cycle with uncertainties that are in the range of those known from the land biota.

## 1 Introduction

With global warming, phytoplankton abundance and distribution are predicted to change but how these changes affect biogeophysical mechanisms such as phytoplankton light absorption remain unknown. Using an Earth system model (ESM) of intermediate complexity, we study the effect of phytoplankton light absorption on the climate system under future climate scenarios.

Observations indicate that the abundance of phytoplankton biomass decreases or will decrease due to global warming (Behren-





feld et al., 2006; Falkowski and Oliver, 2007). For instance, ocean transparency data show that phytoplankton biomass decreases by ∼1% per year since 1899 (Boyce et al., 2010). Furthermore, remotely-sensed ocean color data indicate that between 1998 and 2006, low surface chlorophyll areas have expanded by 15% on a global scale (Polovina et al., 2008). Similar decline

in chlorophyll concentrations is also observed on a regional scale (McClain et al., 2004) and is linked to an increase in sea surface temperature (Gregg et al., 2005). Supporting these observations, most of climate models project a decline in net primary production in the future due to anthropogenic warming. For instance, a CMIP6 model-ensemble study indicates a decrease in depth-integrated primary production of 2.99±9.11% by the end of the 21st century under the high emission scenario SSP5-8.5 (Kwiatkowski et al., 2020). Additionally, using a coupled ocean-biogeochemistry model, Couespel et al. (2021) highlight a

decrease in net primary production of 12% after a linear increase in atmospheric temperature reaching +2.8°C by the end of the 21st century. These changes in phytoplankton abundance, distribution and biogeography have consequently an impact on the role of phytoplankton light absorption.

Different modeling studies investigate the effect of phytoplankton light absorption under global warming. It is speculated

that the decrease in phytoplankton abundance will increase ocean clarity and leads to a lower biological increase of sea surface temperature (SST). A reduction of phytoplankton-induced oceanic warming could thus counteract in part the warming associated with climate change (Patara et al., 2012). To study the effect of phytoplankton light absorption in a warming scenario, Sonntag (2013) conducts simulations with a one-dimensional coupled ocean-biological model. The author runs simulations using the present-day SST and simulations using a homogeneous SST increase of 3°C. Taking into account the biogeophysical

mechanism, Sonntag (2013) detects a local SST increase of 0.2°C in the global warming scenario compared to the present-day scenario. Furthermore, Paulsen (2018) uses another approach where the author performs simulations with and without phytoplankton light absorption under a transient increase of 1% of atmospheric $CO_2$ per year. With phytoplankton light absorption, Paulsen (2018) reports a decline of chlorophyll concentrations associated with a local oceanic warming of up to 0.7°C. This maximum warming is attributed to changes in oceanic circulation, under the global warming scenario. Additionally, using a

coupled ocean-atmosphere model, Park et al. (2015) focus on the Arctic region to study phytoplankton light absorption under global warming. They conduct simulations, with and without phytoplankton light absorption, where atmospheric $CO_2$ concentration increases by 1% per year from the level of 1990 to double its initial concentration. The authors show that phyoplankton light absorption amplifies future Arctic warming by 20%.

So far, the impact of phytoplankton light absorption under oceanic warming (Sonntag, 2013), constant atmospheric $CO_2$

concentration (Patara et al., 2012) and prescribed rising atmospheric $CO_2$ concentrations (Park et al., 2015; Paulsen, 2018) is investigated. However, Asselot et al. (2021b) evidence that phytoplankton light absorption mainly affects the climate system via air-sea $CO_2$ exchange. Thus, prescribing atmospheric $CO_2$ concentrations for global warming simulations blurs the real effect of this biogeophysical mechanism. As a consequence, rather than prescribing the atmospheric $CO_2$ concentrations, we are interested in the effects of phytoplankton light absorption under future $CO_2$ emissions on a long timescale. To address

this question we apply the EcoGEnIE Earth system model (Ward et al., 2018) and force the atmosphere with $CO_2$ emissions





following the four Representative Concentration Pathways (RCP) scenarios used by the Intergovernmental Panel on Climate Change (IPCC) for their Fifth Assessment Report (Meinshausen et al., 2011; IPCC, 2014).

## 2 Methods

### 2.1 The Representative Concentration Pathways scenarios

The RCP scenarios are predicted greenhouse gas emissions and concentration trajectories adopted by the IPCC (Meinshausen et al., 2011; IPCC, 2014). These scenarios describe possible future climate systems depending on the volume of greenhouse gases emitted in the next years (Figure 1). Originally, there are four RCP scenarios, namely RCP2.6, RCP4.5, RCP6.0 and RCP8.5, labeled after a possible radiative forcing values at the beginning of the $22^{nd}$ century (2.6, 4.5, 6.0 and 8.5 W/m$^2$, respectively). These scenarios are consistent with socio-economic assumptions and associated greenhouse gas emissions. They comprise a stringent mitigation scenario (RCP2.6), two intermediate scenarios (RCP4.5 and RCP6.0) and a high greenhouse gas emissions scenario (RCP8.5). This study is conducted on a multi-century timescale to understand the long term climate response and to be able to reach a steady state of our simulations. The RCP scenarios only span the period from 2005 to 2100, a study on multi-century climate analysis, however, requires data beyond 2100. We therefore use the Extended Concentration Pathways (ECPs) designed by stakeholders and scientific groups (Meinshausen et al., 2011). The ECPs assume a constant atmospheric CO$_2$ emissions between 2250 and 2500. Similar to RCP2.6, the ECP2.6 represents a strong mitigation scenario including negative CO$_2$ emissions from 2080 to 2500. For practical purposes, here, referring to the RCP scenarios indicate the period between 1765 and 2500.

### 2.2 Model description

The Earth system mode (ESM)l used in this study is called EcoGEnIE (Ward et al., 2018) and is an association between a new ecosystem component (ECOGEM) and a previous model named cGEnIE (Lenton et al., 2007). EcoGEnIE is an ESM of intermediate complexity (EMIC) (Claussen et al., 2002) and due to the limitation of such a model, we focus on the qualitative assessments rather than on quantitative estimates of our results. Moreover, cGEnIE is widely used to study past climate systems and the carbon cycle over geological timescales (Gibbs et al., 2016; Meyer et al., 2016; Greene et al., 2019; Stockey et al., 2021). EcoGEnIE is already used to analyze the role of marine phytoplankton in the warm early Eocene period (Wilson et al., 2018). This model is also employed to explore the relationships between plankton size, trophic complexity and the availability of phosphorus during the late Cryogenian (Reinhard et al., 2020). We use the same configuration as described in Asselot et al. (2021a) and therefore only briefly explain the climate modules here. This model contains components related to climate processes, including ocean dynamics, marine biogeochemistry, marine ecosystem, atmospheric circulation and sea-ice dynamics (Figure 2). We do not consider a dynamical land scheme, thus the surface land temperature is equal to the surface atmospheric temperature. For this study, we modify the ecosystem component and the oceanic component to implement phytoplankton light absorption.





### 2.2.1 Ocean, atmosphere and sea-ice representation

The oceanic component is a 3D frictional-geostrophic oceanic component (GOLDSTEIN) that calculates the horizontal and
vertical redistribution of heat, salinity and biogeochemical elements (Edwards and Marsh, 2005). The horizontal grid (36 ×
36) is uniform in longitude and uniform in sine latitude, giving ∼3.2° latitudinal increments at the equator increasing to 19.2°
in the polar regions. This horizontal grid is already employed to investigate the global carbon cycle (Cameron et al., 2005).
Furthermore, we consider 32 vertical oceanic layers, increasing logarithmically from 29.38 m for the surface layer to 456.56
m for the deepest layer.

The atmospheric component (EMBM) is closely based on the UVic Earth system model (Weaver et al., 2001). It is a 2D model,
where the atmospheric temperature and specific humidity are the prognostic variables. Precipitation instantaneously removes
all moisture corresponding to the excess above a relative humidity threshold.

The sea-ice component (GOLDSTEINSEAICE) considers ice thickness and ice areal fraction as prognostic variables. The
transport of sea-ice includes the sources and sinks of these variables. Both growth and decay of sea-ice depend on the net heat
flux into the sea-ice. The sea-ice component acts as a coupling module between the ocean and the atmosphere, where heat and
freshwater are exchanged and conserved between these three modules.

### 2.2.2 Ocean biogeochemistry component

The biogeochemical module (BIOGEM) represents the transformation and spatial redistribution of biogeochemical tracers
(Ridgwell et al., 2007). The state variables are inorganic nutrients and organic matter. Organic matter is partitioned into dis-
solved and particulate organic matter (DOM and POM). The model includes iron (Fe) and phosphate ($PO_4$) as limiting nutrients
but similar to Asselot et al. (2021a), we do not consider nitrate ($NO_3^-$) here. Furthermore, BIOGEM calculates the air-sea $CO_2$
and $O_2$ exchange.

### 2.2.3 Ecosystem community component

The marine ecosystem component (ECOGEM) represents the marine plankton community and associated interactions within
the ecosystem (Ward et al., 2018). The biological uptake in ECOGEM is limited by light, temperature and nutrient availability.
Plankton biomass and organic matter are subject to processes such as resource competition and grazing before being passed
to DOM and POM. The ecosystem is divided into different plankton functional types (PFTs) with specific traits. Furthermore,
each PFT is sub-divided into size classes with specific size-dependent traits. We incorporate two classes of PFTs: phytoplankton
and zooplankton. Phytoplankton is characterized by nutrient uptake and photosynthesis whereas zooplankton is characterized
by predation traits. Zooplankton grazing depends on the concentration of prey biomass and prey size. Predominantly zooplank-
ton graze on preys that are 10 times smaller than themselves. Each population is associated with biomass state variables for
carbon, phosphate and chlorophyll. The production of dead organic matter is a function of mortality and messy feeding, with
partitioning between non-sinking DOM and sinking POM. Finally, plankton mortality is reduced at very low biomass such that
plankton cannot become extinct.





### 2.2.4 Phytoplankton light absorption

In a previous study (Asselot et al., 2021a), phytoplankton light absorption has been implemented into ECOGEM and the same approach is used here. In the model configuration of this study, the incoming shortwave radiation varies seasonally. The presence of organic matter, inorganic matter and dissolved molecules limits the propagation of light within the ocean (Ward et al., 2018). The vertical light attenuation scheme is given by Eq. 1

$$I(z) = I_0 \cdot \exp(-k_w \cdot z - k_{Chl} \cdot \int_0^z Chl(z) \cdot dz) \tag{1}$$

where $I(z)$ is the radiation at depth $z$, $I_0$ is the radiation at the surface of the ocean, $k_w$ is the light absorption by clear water (0.04 m$^{-1}$), $k_{Chl}$ is the light absorption by chlorophyll (0.03 m$^{-1}$(mg Chl)$^{-1}$) and $Chl(z)$ is the chlorophyll concentration at depth $z$. The values for $k_w$ and $k_{Chl}$ are taken from Ward et al. (2018). The parameter $I_0$ is negative in the model because it is a downward flux from the sun to the surface of the ocean. The light penetrates until the sixth oceanic layer of the model (221.84 m), with maximum light absorption in the surface layer and minimum absorption in the sixth layer. As demonstrated by e.g. Wetzel et al. (2006); Anderson et al. (2007); Sonntag (2013), phytoplankton changes the optical properties of the ocean through phytoplankton light absorption. It can cause a radiative heating and change the heat distribution in the water column. We implemented phytoplankton light absorption into the model following Hense (2007) and Patara et al. (2012) and the scheme is given by Eq. 2:

$$\frac{\partial T}{\partial t} = \frac{1}{\rho \cdot c_p} \frac{\partial I}{\partial z} \tag{2}$$

$\partial T/\partial t$ denotes the temperature changes, $c_p$ is the specific heat capacity of water, $\rho$ is the ocean density, $I$ is the solar radiation incident at the ocean surface, and $z$ is depth. We assume that the whole light absorption heats the water (Lewis et al., 1983).

### 2.3 Model setup and simulations

We use the same model setup and parametrization as described in (Asselot et al., 2021a), with a 32 layer vertical oceanic grid, primary production allowed until the sixth grid layer (221.84 m deep) and a seasonally variable incoming shortwave radiation. The ecosystem community is consistent with the community described in Asselot et al. (2021a), with one phytoplankton species and one zooplankton species (Appendix A1). We first run a 10,000 years spin-up with only BIOGEM to have a realistic distribution of nutrients. The spin-up is run with a constant pre-industrial atmospheric CO$_2$ concentration of 278 ppm. Then ECOGEM is switched on and the simulations restart after the spin-up for 737 years-long, since the CO$_2$ emissions data are only available for this time span (Meinshausen et al., 2011). We run the simulations with prescribed global CO$_2$ emissions, which are the sum of the fossil, industrial and land-use related CO$_2$ emissions (Figure 1). Moreover, all simulations include ECOGEM and are forced with the same constant flux of dissolved iron into the ocean surface (Mahowald et al., 2006). In total,



we run 8 simulations following the RCP scenarios with and without phytoplankton light absorption (Table 1). We compare the yearly-averaged outputs of the year 2500, when the climate system is in steady state.

## 3  Model validation

To validate our model setup, we compare our results with the results of an EMIC intercomparion (Zickfeld et al., 2013), which has a model setup close to our model setup. We compute the surface atmospheric temperature (SAT) difference between our RCP simulations and a pre-industrial simulation (Appendix B1) without phytoplankton light absorption. In the pre-industrial simulation, the atmospheric $CO_2$ concentration is constant and set to 278 ppm. Independent of the RCP scenario, Figure 3 shows that our increases in SAT are in agreement with Zickfeld et al. (2013). Thus, our model setup is suitable to study the effect of phytoplankton light absorption under future $CO_2$ emissions.

## 4  Results

In this section, we are interested in resolving the effects of phytoplankton light absorption and the relative differences between the simulations. Due to the limitations of such an EMIC, the absolute values are less relevant. In a first time, we look at the ocean properties such as surface chlorophyll biomass and SST. Second, we investigate the changes in atmospheric $CO_2$ concentrations and SAT.

### 4.1  Oceanic properties

#### 4.1.1  Surface chlorophyll biomass

We look at the distribution of chlorophyll at the ocean surface because this climate variable directly affects the heat distribution along the water column and therefore phytoplankton light absorption. On a global scale, independently of the RCP scenario, phytoplankton light absorption leads to an increase of chlorophyll biomass at the ocean surface (Figure 4). For the RCP2.6, RCP4.5 and RCP6.0 scenarios, the global increase of chlorophyll biomass is between 0.015 and 0.019 mgChl/m$^3$, representing an increase between 13 and 15%. These assessments are slightly higher than previous estimates showing an increase between 4 and 12% (Manizza et al., 2005; Asselot et al., 2021a). However, compared to our model setup, Manizza et al. (2005) use an ocean model, neglecting any interactions between the ocean and the atmosphere. Additionally, Asselot et al. (2021a) do not prescribe $CO_2$ emissions, neglecting the changes in chlorophyll biomass due to climate change. The increase of chlorophyll biomass for the RCP8.5 scenario is the smallest, with an increase of ~0.01 mgChl/m$^3$, representing an increase of 8%. The regional patterns of surface chlorophyll biomass changes due to phytoplankton light absorption are similar between the RCP scenarios (Figure 5). The largest differences of chlorophyll biomass occur in the high latitudes. Such as, between the simulations *RCP8.5-LA* and *RCP8.5*, the maximum increase of 0.4 mgChl/m$^3$ occurs in the northern polar region (Figure 5d). This pronounced chlorophyll increase is due to the coarse grid resolution and the global decrease of sea-ice in the polar region. For instance, the global sea-ice cover decreases by 0.21% between *RCP8.5-LA* and *RCP8.5*, thus increasing the light availability





for phytoplankton growth. In addition, the upwelling and mid-latitude regions evidence a higher chlorophyll biomass with phytoplankton light absorption in opposition to the subtropical gyres, where no or small differences occur. These regional pat-

180  terns of higher chlorophyll biomass are due to an enhanced vertical velocity which is triggered by changes in the oceanic heat budget (see below). For instance, in the upwelling region along the western African coast, at 326 m depth, the vertical velocity is enhanced by 9.2% in *RCP4.5-LA* compared to *RCP4.5*. As a consequence, on a global scale, more nutrients are brought to the surface, decreasing the nutrient limitation and thus promoting a higher phytoplankton biomass at the surface.

### 4.1.2  Sea surface temperature

185  Due to changes in surface chlorophyll biomass, we expect variations in sea surface temperature. Because of our model setup, the SST is the zonally-averaged temperature from the surface to 29 m depth. Our results highlight that under the RCP2.6, RCP4.5 and RCP6.0 scenarios, phytoplankton light absorption increases the SST by ∼0.6°C. These assessments are higher than previous global estimates indicating a global SST increase of 0.33-0.5°C (Wetzel et al., 2006; Patara et al., 2012; Asselot et al., 2021a). This stronger increase in SST is caused by higher increases in surface chlorophyll biomass compared to previous

190  assessments. For the RCP8.5 scenario, phytoplankton light absorption only increases the SST by 0.23°C. This lower increase in SST is due to the lower global increase in surface chlorophyll biomass for this scenario. The regional patterns of SST changes due to phytoplankton light absorption are similar between the simulations but the magnitude of changes differs (Figure 6). Independently of the RCP scenario, the polar regions experience the lowest increase of SST due to the coarse resolution and the sea-ice dynamics limiting the heat redistribution. For instance, between the simulations *RCP4.5-LA* and *RCP4.5*, the

195  minimum increase of 0.03°C occurs in the Southern Ocean. Even in the regions where small differences in surface chlorophyll occur, such as the subtropical gyres, we find high SST increases. The missing spatial patterns between chlorophyll biomass and SST can be explained by the model setup. The state variables of the ecosystem component, such as surface chlorophyll biomass, are not subject to transport while physical quantities, such as heat, are transported by ocean currents. Therefore, heat is smoothly redistributed around the globe.

## 4.2  Atmospheric properties

Since the oceanic properties, specifically chlorophyll biomass and SST, change due to phytoplankton light absorption, the atmospheric properties are also expected to change.

### 4.2.1  Atmospheric $CO_2$ concentration

Even if we prescribe RCP emissions in our simulations, the expected atmospheric $CO_2$ concentrations in 2500 under the RCP

205  scenarios are not reached. With primary production allowed until the sixth oceanic layer, EcoGEnIE has not been tuned yet. Thus, with this configuration, the model is known to simulate a low atmospheric $CO_2$ concentration (Asselot et al., 2021b, a). Because we are more interested in qualitative assessment rather than quantitative estimates, such limitation does not affect the main findings of our study. The atmospheric $CO_2$ concentrations are the lowest in the simulations following the RCP2.6





scenario due to the lowest and negative emissions under this scenario. In contrast, the atmospheric $CO_2$ concentrations are the highest for the simulations following the RCP8.5 scenario due to the highest emissions in this scenario. Independently of the RCP scenario, the atmospheric $CO_2$ concentration increases with phytoplankton light absorption (Figure 7). For the RCP2.6, RCP4.5 and RCP6.0 scenarios, phytoplankton light absorption increases the atmospheric $CO_2$ concentration by ∼20% while a previous study indicates an increase of 10% (Asselot et al., 2021a). However, in Asselot et al. (2021a) we do not prescribe any $CO_2$ emissions, neglecting their effect on the atmospheric $CO_2$ concentration. Additionally, for the RCP8.5 scenario, the atmospheric $CO_2$ concentration increases only by 8% due to the lower changes in oceanic properties.

### 4.2.2 Surface atmospheric temperature

The global changes in oceanic and atmospheric properties due to phytoplankton light absorption lead to an increase of surface atmospheric temperature (SAT) (Figure 8). For the RCP2.6, RCP4.5 and RCP6.0 scenarios, the global increase in SAT due to phytoplankton light absorption is ∼0.8°C. This value is higher than previous model studies estimating a zonally-averaged SAT increase of 0.2-0.45°C (Shell et al., 2003; Patara et al., 2012; Asselot et al., 2021a). However, compared to our model setup, Shell et al. (2003) use an uncoupled ocean-atmosphere model, neglecting any interactions between the ocean and the atmosphere. Patara et al. (2012) use a constant and prescribed atmospheric $CO_2$ concentration for their simulations, neglecting its effect on the atmospheric temperature. Asselot et al. (2021a) do not prescribe $CO_2$ emissions, neglecting changes in the heat budget due to climate change. With a value of 0.28°C, the increase in SAT under the RCP8.5 scenario is lower than for the other RCP scenarios. The regional patterns of SAT changes due to phytoplankton light absorption are similar among the RCP scenarios but the magnitude of changes differs (Figure 9). The polar regions experience a strong increase in SAT, with the highest values occurring in the Southern Ocean. For instance, comparing the simulations *RCP4.5-LA* and *RCP4.5*, the maximum increase of 1.6°C occurs in the Southern Ocean (Figure 9b). This maximum value might be due to the rather coarse grid resolution in the high latitudes. This estimate is again higher than previous local estimates of model studies for the same reasons described above (Shell et al., 2003; Patara et al., 2012; Asselot et al., 2021a). Furthermore, along the rest of the globe, the heat is redistributed smoothly.

### 4.3 Synthesis

Under climate change scenarios we find an increase in surface chlorophyll biomass due to two mechanisms. First, phytoplankton light absorption leads to a higher surface production, increasing the remineralization and thus the nutrient concentrations at the ocean surface. Second, this biogeophysical mechanisms enhances the upward vertical velocity, especially in the upwelling regions, trapping more nutrients at the ocean surface. The higher surface nutrient concentrations, specifically the phosphate concentrations (Appendix C1), resulting from these two mechanisms explain the higher surface chlorophyll biomass with phytoplankton light absorption. Via the effect of this biogeophysical mechanism, a higher surface chlorophyll biomass leads to more heat being trapped in the ocean surface. As a result, the SST increases with phytoplankton light absorption (Figure 4). Due to the physical and chemical properties of the ocean, the higher SST decreases the solubility of gases such as $CO_2$. Similar to Asselot et al. (2021b), the sea-air $CO_2$ flux is thus enhanced with phytoplankton light absorption, increasing the atmospheric





CO$_2$ concentrations (Figure 7). Via the greenhouse gas effect, the higher atmospheric CO$_2$ concentrations lead to higher SATs with phytoplankton light absorption (Figure 8).

## 5    Discussion

### 5.1    General discussion

Our results evidence that under the RCP2.6, RCP4.5 and RCP6.0 scenarios, the impact of phytoplankton light absorption on the climate system is of the same order of magnitude. This is due to the similar metabolic processes that favour the growth of phytoplankton for these simulations. However, under the RCP8.5 scenario, the effect of phytoplankton light absorption on the climate system is reduced. This is due to the model setup where a SST higher than 20°C limits phytoplankton growth

(Ward et al., 2018). This threshold is only exceeded for the the simulations *RCP8.5-LA* and *RCP8.5* (Appendix D1), therefore phytoplankton growth is limited by the ocean temperature in these two simulations (Figure 10). As a result, the difference of chlorophyll biomass between *RCP8.5-LA* and *RCP8.5* is weaker than between the other simulations (Figure 4). The weaker increase in surface chlorophyll biomass explains the weaker response of the climate system to phytoplankton light absorption under the RCP8.5 scenario compared to the other RCPs scenarios.


For the simulations following the RCP2.6 scenario, the final atmospheric CO$_2$ concentrations and SSTs are lower than the pre-industrial levels (Appendix B1). This is due to the negative emissions for this scenario and the underestimation of the atmospheric CO$_2$ concentrations with our model setup (Asselot et al., 2021b, a). As detailed previously, primary production is allowed until the sixth oceanic layer and the model has not been tuned in this configuration yet. The lower levels under the

RCP2.6 scenario compared to the pre-industrial levels are not an issue for our study because we exclusively focused on the effect of phytoplankton light absorption rather than on the differences between the simulations and the pre-industrial state.

### 5.2    Implication for Earth system models

The traditional view is that dominant carbon cycle uncertainties come from the terrestrial response to elevated atmospheric CO$_2$ concentrations. For instance, the net land emissions over the post-industrial period is estimated to lie in the range from

0 to 128 GtC (Holden et al., 2013). However, this work suggests that introducing biogeophysical mechanisms such as phytoplankton light absorption leads to major carbon cycle uncertainties. For instance, with our model setup and under the RCP2.6 scenario, implementing phytoplankton light absorption increases the atmospheric carbon content by 79 GtC. This study highlights a highly uncertain feedback on the carbon cycle that is missing from most Earth system models used for the CMIP5 and CMIP6 projects (Pellerin et al., 2020). Neglecting the effect of phytoplankton light absorption on the carbon cycle can lead to

incomplete future climate predictions.





## 6 Conclusions

Phytoplankton light absorption is previously investigated under constant $CO_2$ forcing (Patara et al., 2012), transient atmospheric $CO_2$ increase (Park et al., 2015; Paulsen, 2018) and under artificial SST increase (Sonntag, 2013). For the first time, using the EcoGEnIE model (Ward et al., 2018), we investigate the impact of phytoplankton light absorption under prescribed
$CO_2$ emissions following the RCP scenarios on a multi-century timescale.

Independently of the RCP scenario, our results suggest that phytoplankton light absorption causes an increase in surface chlorophyll biomass, SST, atmospheric $CO_2$ concentration and SAT. For the RCP2.6, RCP4.5 and RCP6.0 scenarios, the responses of the climate system due to phytoplankton light absorption are of the same order of magnitude. Under the RCP2.6,
RCP4.5 and RCP6.0 scenarios, the oceanic conditions favor phytoplankton growth, leading to a strong effect of phytoplankton light absorption. However, for the RCP8.5 scenario, the changes in the climate system due to phytoplankton light absorption are lower than for the other RCP scenarios. Under the RCP8.5 scenario, the strong warming of the ocean is not as favorable as for the previous RCP scenarios, limiting phytoplankton growth and thus reducing the effect of phytoplankton light absorption. This result indicates that the effect of phytoplankton light absorption is smaller under a high greenhouse gas emissions com-
pared to reduced and intermediate greenhouse gas emissions. In agreement with Patara et al. (2012), our findings indicate that a severely warmer world increases the ocean clarity and slows down the phytoplankton-induced global warming. Moreover, this work highlights the major role of phytoplankton light absorption on the climate system, suggesting a highly uncertain feedback on the carbon cycle with uncertainties in the atmospheric content that are in the range of those known from the land biota. Neglecting the link between phytoplankton light absorption and the carbon cycle could thus alter the future climate simulations.

Our study is designed to explain the processes behind the future impact of phytoplankton light absorption on the climate system but our model setup has limitations. To improve our quantitative estimates, such limitations must be overcome. Most notably, the model must be tuned to fit the expected atmospheric $CO_2$ concentrations under global warming scenarios. Additionally, more observations and research are needed to understand and simulate the role of phytoplankton in a changing climate
system. Specifically, microorganisms such as cyanobacteria are expected to expand with future climate change (O'neil et al., 2012; Paerl and Paul, 2012; Ullah et al., 2018) and their impact on the climate system through phytoplankton light absorption is large (Anderson et al., 2007; Sonntag, 2013; Paulsen et al., 2018). A logical follow up would be to include cyanobacteria in a similar study. Additionally, our model setup does not consider phytoplankton adaptation to higher temperature. The reduced effect of phytoplankton light absorption under the RCP8.5 scenario may not be as strong if phytoplankton adapts to higher
temperature. Implementing phytoplankton adaptation would be an interesting extension to our work.



*Code availability.* The code for the model is hosted on GitHub and can be obtained by cloning or downloading: https://zenodo.org/record/5676165. The configuration file is named "RA.ECO.ra32lv.FeTDTL.36x36x32" and can be found in the directory "EcoGENIE_LA/genie-main/configs". The user-configuration files to run the experiments can be found in the directory "EcoGENIE_LA/genie-userconfigs/RA/Asselotetal_ESD". Details of the code installation and basic model configuration can be found on a PDF file (https://www.seao2.info/cgenie/docs/muffin.pdf).
Finally, section 9 of the manual provides tutorials on the ECOGEM ecosystem model.

## Appendix A: Plankton functional types

We base our ecosystem community on the one described by Ward et al. (2018). We only use 2 PFTs: one phytoplankton group and one zooplankton group (Appendix A1). We show that the complexity of the ecosystem does not have an important impact on the climate system compared to the effect of phytoplankton light absorption (Asselot et al., 2021a). Therefore, for 310 simplification, we reduce the ecosystem complexity.

## Appendix B: Pre-industrial scenario

For the simulations under the RCP2.6 scenario, the atmospheric $CO_2$ concentrations and SSTs are lower than the pre-industrial levels. This is due to the negative emissions for this scenario and the underestimation of the atmospheric $CO_2$ concentration with our model configuration.

## Appendix C: Surface phosphate concentration

Independent of the RCP scenarios, our results evidence an increase in surface nutrients, such as phosphate. As a result, the surface chlorophyll biomass increases with phytoplankton light absorption.

## Appendix D: Sea surface temperature

Following Ward et al. (2018), the SST is limiting if it is higher than 20°C. Our results indicate that SST is limiting phytoplank-
ton growth for the simulations following the RCP8.5 scenario.

*Author contributions.* All authors designed and developed the concept of the study. RA performed the analysis of the model outputs with inputs from IH. RA drafted the initial version of the manuscript in collaboration with IH. All co-authors read and reviewed the final version of the manuscript.

*Competing interests.* The authors declare that they have no conflict of interest.



*Acknowledgements.*  Our special thanks go to Jana Hinners, Isabell Hochfeld, Félix Pellerin, Maike Scheffold and Laurin Steidle for their valuable comments on the early version of this manuscript. This work was supported by the Center for Earth System Research and Sustainability (CEN), University of Hamburg, and contributes to the Cluster of Excellence "CLICCS - Climate, Climatic Change, and Society".





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





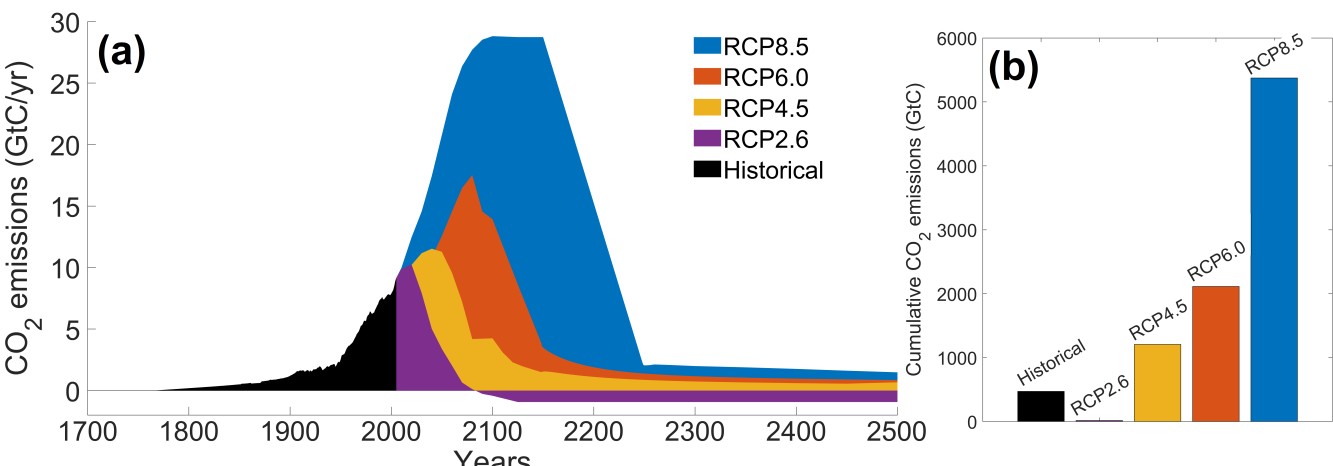

**Figure 1.** Atmospheric $CO_2$ emissions following the Representatives Concentrations Pathways scenarios. (a) Historical and predicted $CO_2$ emissions over time (GtC/yr). (b) Cumulative $CO_2$ emissions for the different scenarios (GtC). The historical emissions represent the cumulative $CO_2$ emissions from 1765 to 2005. The RCP scenarios represent the cumulative $CO_2$ emissions between 2006 and 2500. The color coding between the two panels is similar.





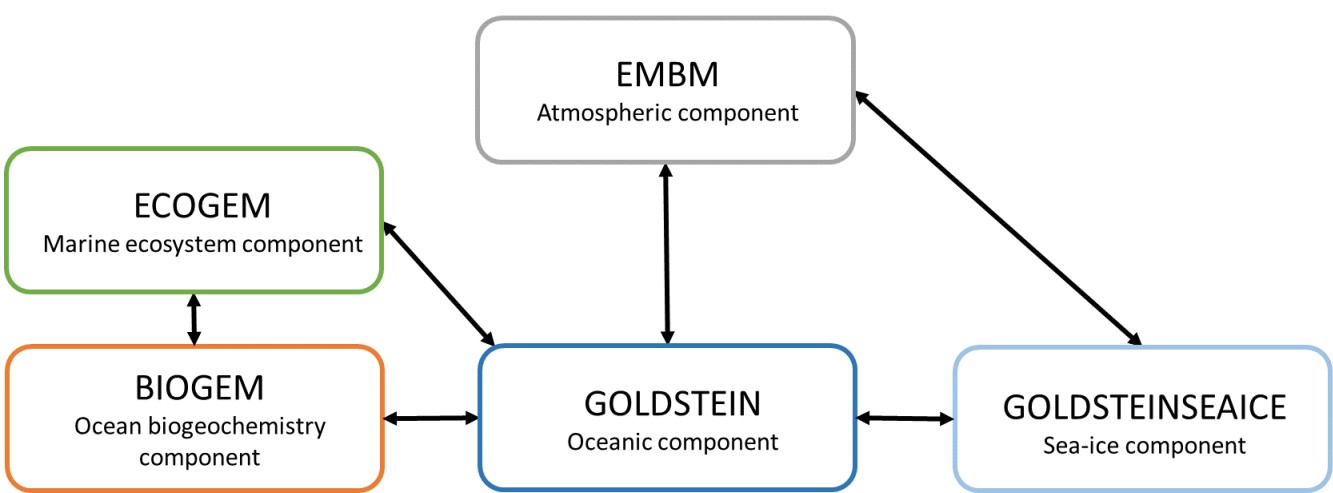

**Figure 2.** Sketch representing the different components of the EcoGEnIE model. Black arrows represent the links between the different components. Figure from Asselot et al. (2021b).





**Figure 3.** Difference in SAT (°C) for our study and the study of Zickfeld et al. (2013). The SAT changes of our study come from simulations without phytoplankton light absorption.





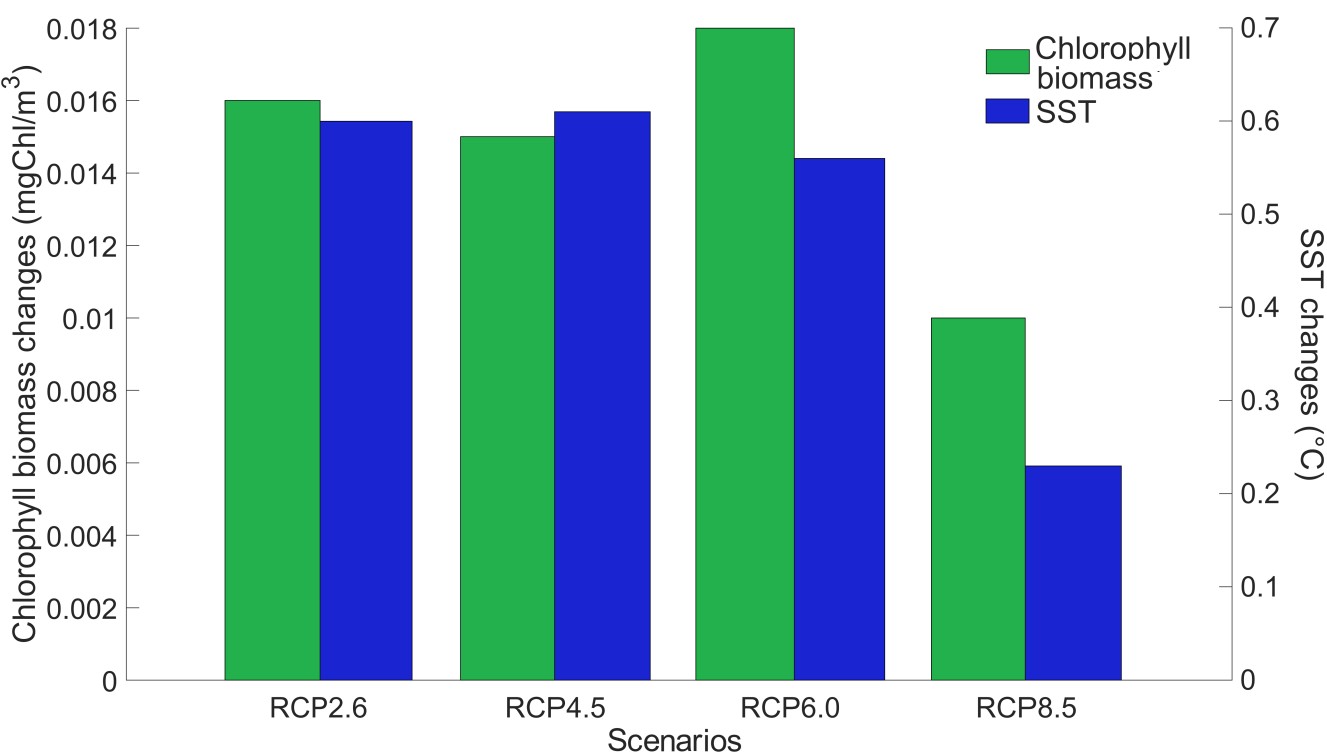

**Figure 4.** Zonally averaged surface chlorophyll biomass (mgChl/m$^3$) and SST ($^\circ$C) changes between the RCP scenarios. The values represent the difference between the simulation with minus without phytoplankton light absorption. Note that the y-axis scales are always positive, indicating that phytoplankton light absorption always leads to a global increase of surface chlorophyll biomass and SST.

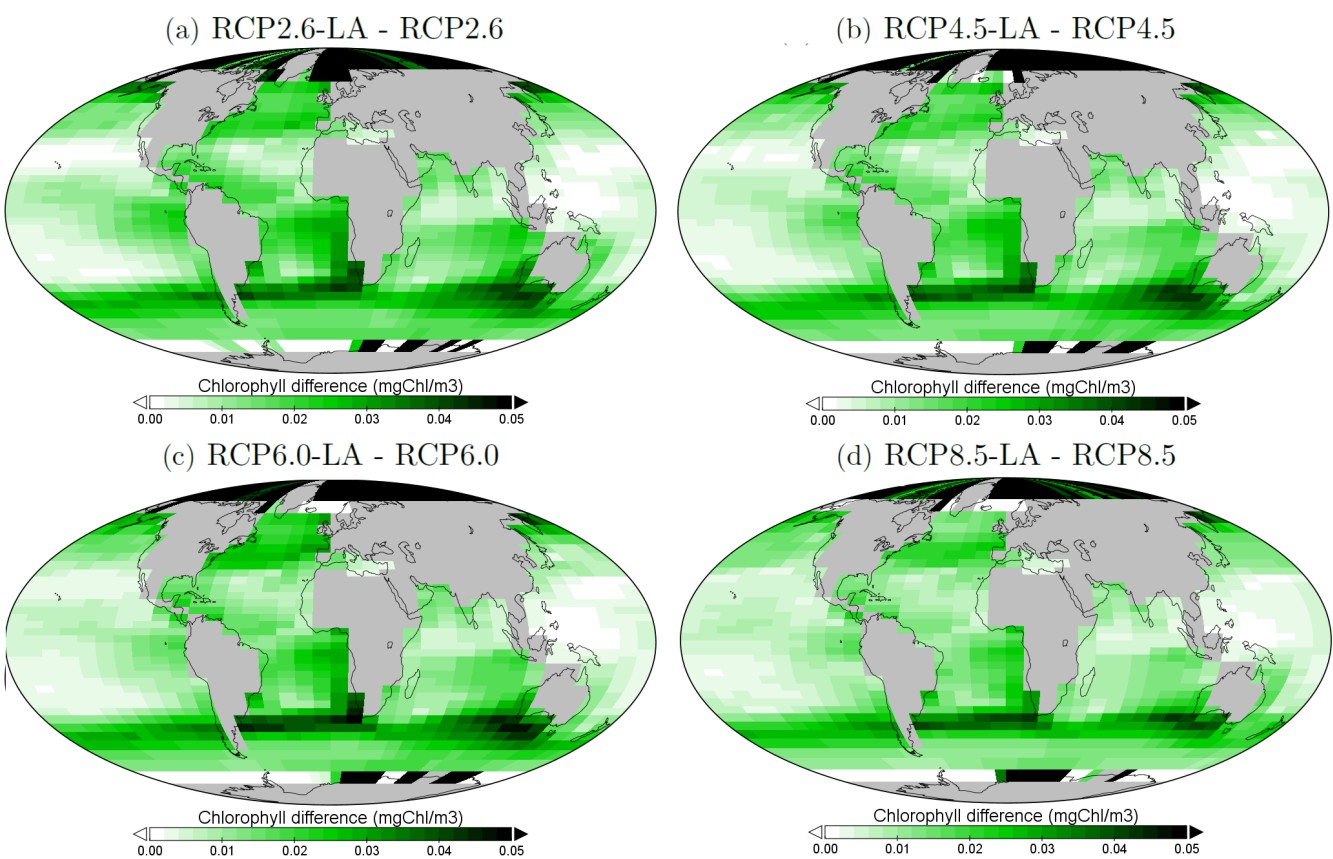

**Figure 5.** Chlorophyll biomass changes at the surface (mgChl/m$^3$) for the different simulations. (a) Difference between RCP2.6-LA and RCP2.6. (b) Difference between RCP4.5-LA and RCP4.5. (c) Difference between RCP6.0-LA and RCP6.0. (d) Difference between RCP8.5-LA and RCP8.5. The scale and color coding are identical between the four panels. Note that the scale is always positive.



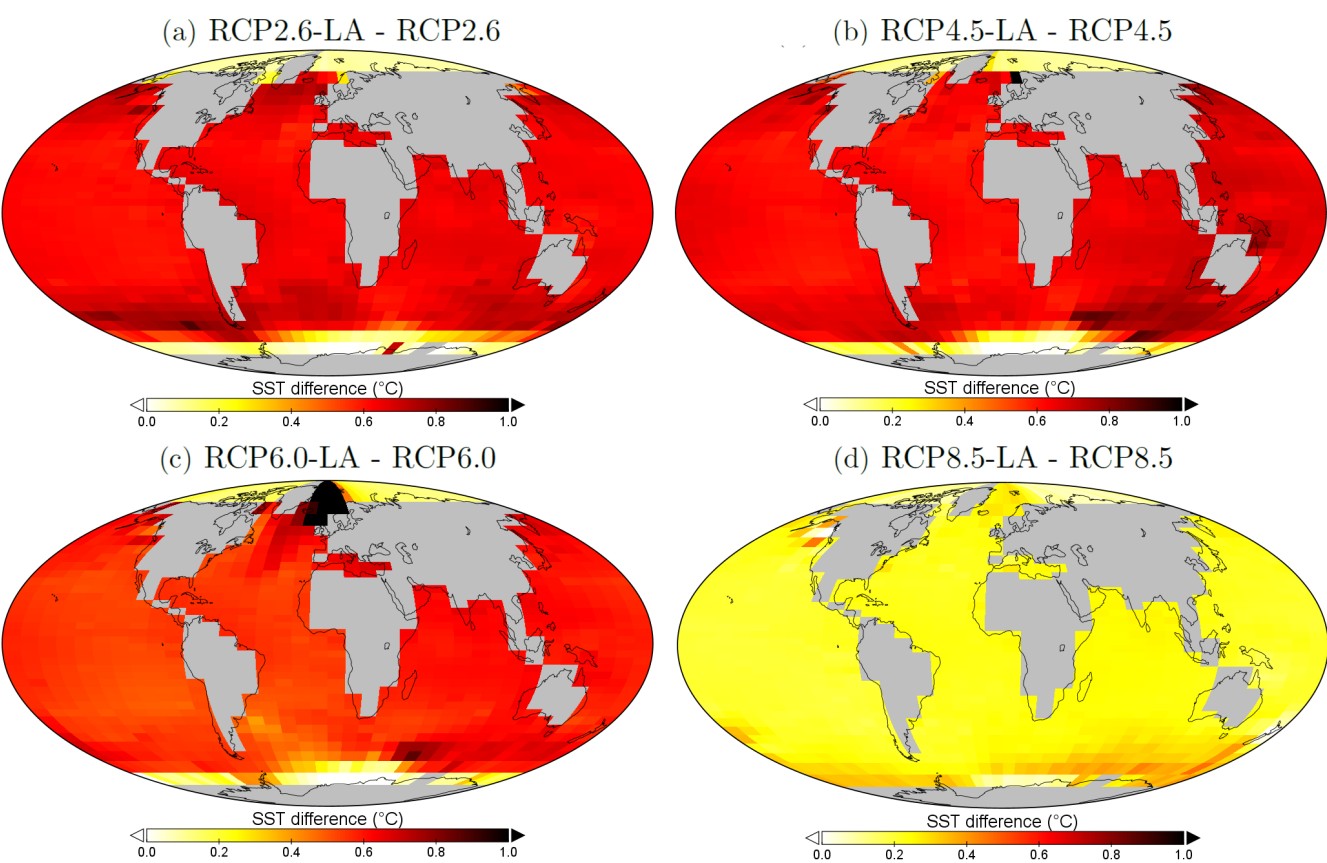

**Figure 6.** Sea surface temperature changes (°C) between the simulations. (a) Difference between RCP2.6-LA and RCP2.6. (b) Difference between RCP4.5-LA and RCP4.5. (c) Difference between RCP6.0-LA and RCP6.0. (d) Difference between RCP8.5-LA and RCP8.5. The scale and color coding are identical between the four panels. Note that the scale is always positive.



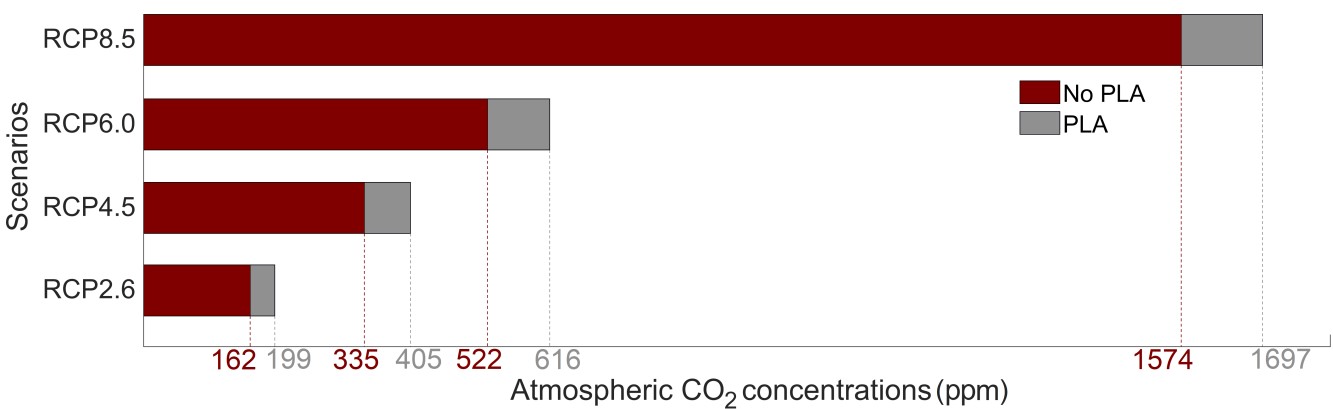

**Figure 7.** Atmospheric $CO_2$ concentrations (ppm) for the 8 simulations. PLA stands for phytoplankton light absorption.



**Figure 8.** Globally-averaged SAT (°C) changes between the RCP scenarios. The values represent the difference between the simulation with and without phytoplankton light absorption. Note that the y-axis scale is always positive, indicating that phytoplankton light absorption always leads to a global increase of SAT.



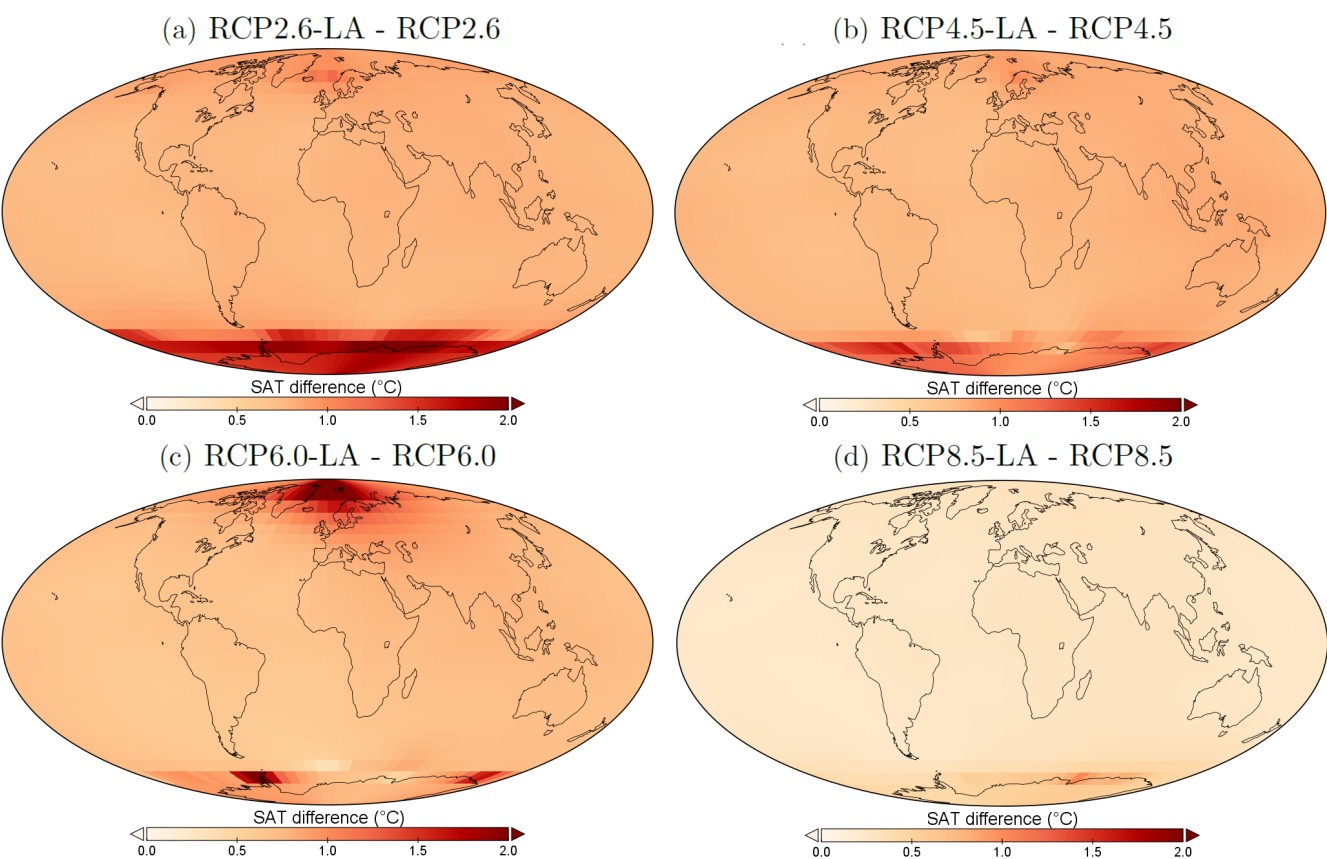

**Figure 9.** Surface atmospheric temperature changes (°C) between the different simulations. (a) Difference between RCP2.6-LA and RCP2.6. (b) Difference between RCP4.5-LA and RCP4.5. (c) Difference between RCP6.0-LA and RCP6.0. (d) Difference between RCP8.5-LA and RCP8.5. The scale and color coding are identical for the four panels. Note that the scale is always positive.



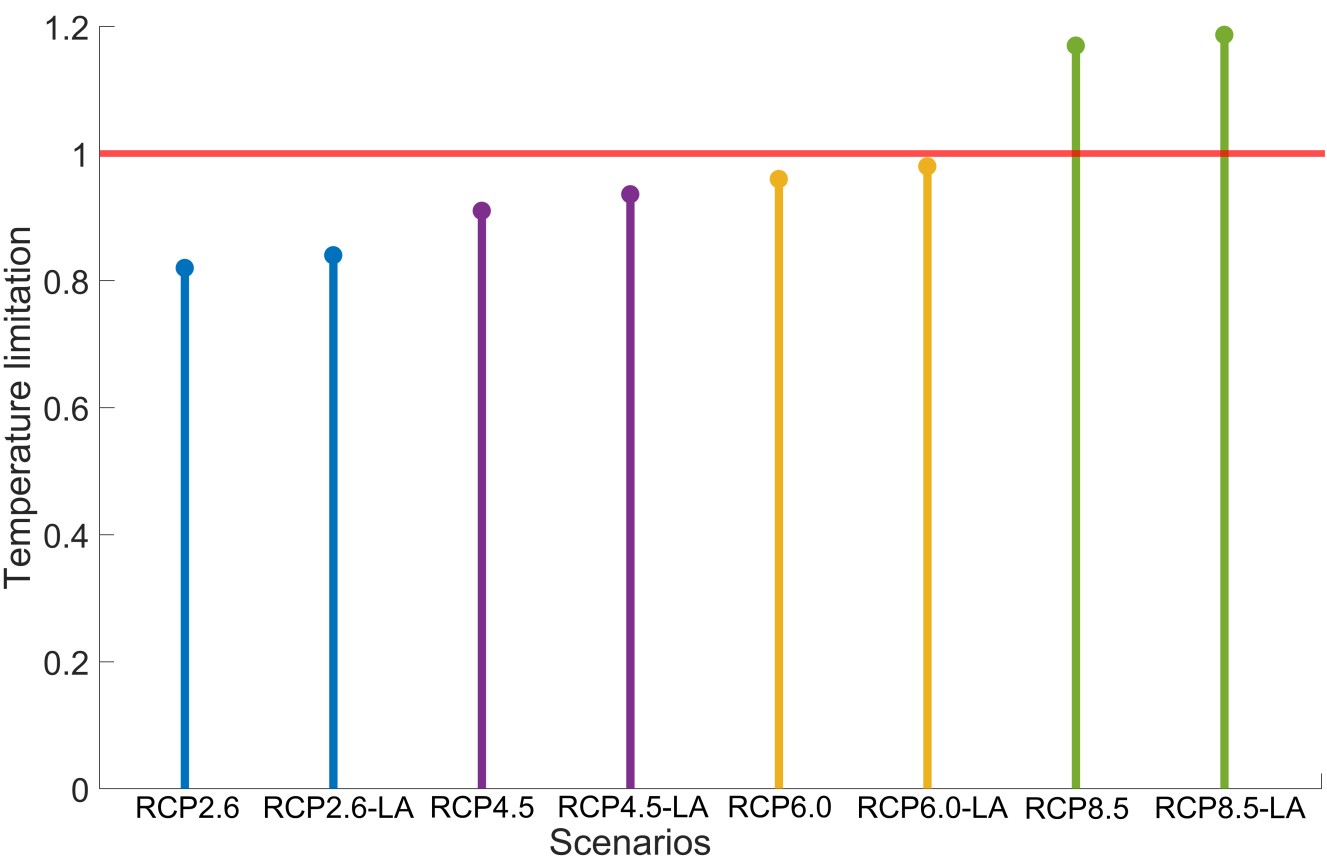

**Figure 10.** Temperature limitation for phytoplankton growth in the 8 simulations. Values over 1 indicate that oceanic temperature is a limiting factor for phytoplankton growth.



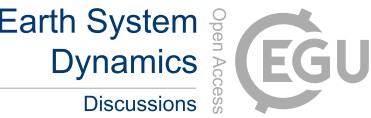

**Table 1.** Name and description of the simulations (PLA = phytoplankton light absorption).

| Name | Description |
| --- | --- |
| RCP2.6 | $CO_2$ emissions following RCP2.6 |
| RCP2.6-LA | $CO_2$ emissions following RCP2.6 with PLA |
| RCP4.5 | $CO_2$ emissions following RCP4.5 |
| RCP4.5-LA | $CO_2$ emissions following RCP4.5 with PLA |
| RCP6.0 | $CO_2$ emissions following RCP6.0 |
| RCP6.0-LA | $CO_2$ emissions following RCP6.0 with PLA |
| RCP8.5 | $CO_2$ emissions following RCP8.5 |
| RCP8.5-LA | $CO_2$ emissions following RCP8.5 with PLA |



**Table A1.** Size of the different plankton functional types ($\mu$m) used during the simulations.

| PFT | Size ($\mu$m) |
|---|---|
| Phytoplankton | 46.25 |
| Zooplankton | 146.15 |



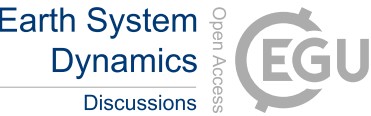

**Table B1.** Comparison of SSTs (°C) and atmospheric $CO_2$ concentrations (ppm) for the simulations under the RCP2.6 scenario and the pre-industrial scenario.

| Simulation | SST (°C) | Atm. $CO_2$ concentration (ppm) |
|---|---|---|
| RCP2.6 | 15.95 | 162 |
| RCP2.6LA | 16.54 | 199 |
| Pre-industrial | 18.57 | 278 |



**Table C1.** Phosphate concentration changes at the surface (mol/kg) for the different RCP scenarios. The values represent the difference with minus without phytoplankton light absorption. The "+" symbol indicates an increase in surface $PO_4$ concentration.

| Scenario | $\Delta PO_4$ conc. (mol/kg) |
|----------|------------------------------|
| RCP2.6   | $+8.9{\cdot}10^{-8}$         |
| RCP4.5   | $+8.6{\cdot}10^{-8}$         |
| RCP6.0   | $+9.1{\cdot}10^{-8}$         |
| RCP8.5   | $+7.5{\cdot}10^{-8}$         |



**Table D1.** Sea surface temperature (°C) for the different simulations.

| Simulations | SST(°C) |
|---|---|
| RCP2.6 | 15.95 |
| RCP2.6LA | 16.54 |
| RCP4.5 | 18.08 |
| RCP4.5LA | 18.68 |
| RCP6.0 | 19.18 |
| RCP6.0LA | 19.75 |
| RCP8.5 | 23.21 |
| RCP8.5LA | 23.44 |