# Peer review of "A missing link in the carbon cycle: phytoplankton light absorption"

_Earth System Dynamics, 2021_

## Referee Comment (RC1)

**Asselot et al. – A missing link in the carbon cycle: phytoplankton light absorption.**

**Summary**

Phytoplankton light absorption is the process by which incident short wave radiation is absorbed by phytoplankton in the ocean's surface waters. This process increases the amount of heat that is absorbed in surface layers while decreasing the depth to which this radiation penetrates. Because phytoplankton also decrease the albedo of the ocean, the net effect is that more shortwave radiation is absorbed in the ocean, specifically the near-surface waters, when more phytoplankton are present.

Asselot et al. present experiments that explore the effect of phytoplankton light absorption on key climate variables of societal importance: ocean surface temperatures, phytoplankton biomass, atmospheric $CO_2$ concentrations, and surface atmospheric temperatures. They do so on the global scale. To do so, they use an Earth System Model of Intermediate Complexity (an ESMIC) called the EcoGEnIE. The ocean-biogeochemical-ecosystem components of the model are equipped with phytoplankton light absorption, such that experiments can be performed with this process turned on or off.

Asselot et al. acknowledge that other studies have already undertaken modelling experiments that assess the effect of phytoplankton light absorption under climate change scenarios. The novelty of their study is therefore in (1) its use of multiple, extended RCP scenarios (2.6, 4.5, 6.0 and 8.5), and (2) its insights into the effect on atmospheric $CO_2$ concentrations since they use emissions, rather than specifying the atmospheric $CO_2$ concentration.

The main findings of this study are:

- That phytoplankton light absorption increases surface temperatures and phytoplankton biomass.
- That the effects are strongest under the weaker global warming scenarios.
- That the warming of the surface ocean weakens the ocean's ability to absorb $CO_2$, which increases atmospheric $CO_2$ and thereby increases surface temperatures in a positive feedback response.

**Major comments**

This paper is presented simply, and in that sense the authors are honest with their results. They do not oversell the lessons. However, this suggests that the mechanisms involved are very simple, and actually I am left wanting more. I thus have three major concerns with this paper as it stands. All relate to the discussion of their results.

First, the authors do not explain the mechanisms (physical and/or biogeochemical) that underly the relatively greater chlorophyll biomass when phytoplankton light absorption is included. As far as I can tell from the work, they speculate about the mechanism, but do not definitively show it. If upwelling increases, why? I think the paper would be strongly improved with a more concrete explanation of this process. I have suggested that the authors undertake some 1D modelling work in my specific comments below, but if they can identify the mechanism with output from the 3D model than that is also fine. Bottom line, the reader needs a clearer more convincing explanation.

Second, the real novelty of this study is in the feedback on $CO_2$ concentration, as the authors state in the introduction. No other modelling study (to theirs and my knowledge) has specifically targeted this feedback. However, the consequences for atmospheric $CO_2$ are reported but the mechanisms are not explained. Why exactly does phytoplankton light absorption increase atmospheric $CO_2$? What I'm really asking here is what portion of the atmospheric $CO_2$ increase can be apportioned to a decrease in the solubility pump, a decrease in the biological pump or a decrease in the carbonate pump (assuming this is included)? I have little doubt that the increase in surface temperature is the main culprit, but quantifying these terms would significantly improve the paper.

Third, the discussion is very short. There is no discussion of which models include this feedback as part of their architecture and which do not. There is little discussion of observational studies that see this in the real ocean, nor a discussion of whether the mechanisms that cause a relative increase in chlorophyll biomass are realistic. There is little discussion regarding how phytoplankton community composition changes might affect the magnitude of phytoplankton light absorption in the future (i.e. cyanobacteria becoming dominant). There is also no acknowledgement of the uncertainty in future primary production, which may increase, remain stable or decrease depending on the model and region of interest. Would phytoplankton light absorption cause substantially greater greenhouse warming if global warming was coincident with stable or increasing global primary production? To what degree would the solubility pump outweigh the oceanic gains in carbon from the biological pump? These sorts of discussion points would be highly beneficial to this paper.

Finally, the appendices are not really appendices. They are single sentence additions, and they really shouldn't be appendices if they can easily be added to the main text.

**Specific comments**

- Line 21: "decreases or will decrease" isn't that the same thing?
- Line 23: Using (Boyce et al. 2010) reference is not recommended. See the responses to this paper in Nature underlining its caveats (https://www.nature.com/articles/nature09951)
- Line 27: -2.99 ± 9.11 is not significantly different from zero. The overall response of the oceanic primary production to global warming is highly uncertain. I would stress that as biogeochemical models have evolved from CMIP5 to CMIP6, the response of phytoplankton biomass and Net Primary Production are more uncertain. This doesn't undermine your study, but it I think it is important to tone down the confidence with which you are projecting a decline.
- Line 43: "… reports a decline of chlorophyll concentrations associated with a local oceanic warming of up to 0.7 °C. This maximum warming is attributed to changes in ocean circulation, under the global warming scenario". This could be written more clearly. I am unsure of what you mean.
- Line 75: typo in model
- Line 144: It is unclear what is done following the spin-up. What years is the 737-year run and which years it is running from? If the run begins at 1765, then your total number of years is 736, not 737.
- Line 150: It is odd to consider your model validation as being a comparison with other models. I would advocate that you either compare your model with observations, or your change the title of this section to "model inter-comparison".
- Line 151: EMIC has not yet been defined.

- Section 4.1.1: Here I am a little dissatisfied with the explanation of why chlorophyll concentrations are greater in the simulation with phytoplankton light absorption. All else being equal, if more shortwave radiation is absorbed in the upper layers of the ocean and less is able to penetrate down to deeper layers, then this should increase the vertical density gradient through the upper water column. An increase in stratification is the result, limiting vertical mixing of nutrients to near-surface layers. However, I suppose that if less radiation penetrated to deeper layers, then the strongest density gradients would exist nearer the surface, lessening density gradients deeper in the water column and thereby increasing the nutrient flux to the lower euphotic zone? All in all, I think I need a clearer explanation of the physical mechanism that is occurring. It would be informative and highly beneficial to this paper to conduct 1D water column simulations that test the effect of phytoplankton light absorption on the nutrient fluxes to phytoplankton.
- Line 205: "not been tuned yet". This suggests that the model has never been tuned. Surely that's not correct.
- Section 4.2.1: What component of the increase in atmospheric $CO_2$ is due to (i) the solubility pump, (ii) the carbonate pump and (iii) the biological pump? Just saying that the increase in $CO_2$ is due to additional surface heating is not convincing. I am not saying that the majority of it is due to biological mechanisms, but surely a portion of the change is due to the changes in phytoplankton biomass being closer to the surface ocean, and therefore not exporting organic carbon to deeper levels as efficiently? Alternatively, the increase in phytoplankton biomass may in fact tend to increase the air-sea flux of $CO_2$, but this is more than opposed by the solubility effect of additional warming.
- Line 233-236: These mechanisms have been proposed to explain to explain the increase in phytoplankton biomass in the runs with phytoplankton light absorption, but they have not been explained mechanistically or shown.
- Line 249: The limitation of phytoplankton growth at temperatures greater than 20 °C is an odd choice… What is the motivation for this choice in parameterisation? Because this parameterisation is really quite arbitrary and probably not realistic (i.e. does not follow the Eppley curve (Eppley 1972) nor the recent work by Anderson et al. (Anderson et al. 2021) that confirms community-wide exponential increases in growth rates with temperature), I have to encourage the authors to discuss how this result for RCP8.5 simulations is likely not realistic either. Also, please tell us what the parameterisation is. What is the equation?
- Line 250: Additionally, the limitation is definitely present in all your scenarios, but RCP8.5 is likely the only one where enough surface grid cells exceed 20 °C that it has a significantly negative effect on the chlorophyll/temperature increase. It is worth mentioning that this is the case. In fact, it may explain the slight decreases in the phytoplankton light absorption effect for each RCP scenario from 2.6 to 4.5 to 6.0.
- Paragraph beginning Line 276: This paragraph discusses the stronger effect of phytoplankton light absorption in RCP scenarios 2.6, 4.5 and 6.0, with RCP8.5 showing a weaker role. The authors take this opportunity to explain that this effect is weaker in RCP8.5 because stronger warming causes an increase clarity (by reducing phytoplankton biomass) and therefore the phytoplankton absorption effect is weaker. But, the authors have only just discussed that the weaker effect in RCP8.5 is due to the arbitrary limitation of phytoplankton above 20 °C. The different response in RCP8.5 is therefore an artefact of the model, and is not realistic. Therefore, it is not evidenced by your study to say that "our findings indicate that a severely warmer world increases ocean clarity and slows down the phytoplankton-induced global warming".

Thank you for considering my input to your research.

Pearse J. Buchanan

**References**

Anderson, S. I., A. D. Barton, S. Clayton, S. Dutkiewicz, and T. A. Rynearson. 2021. Marine phytoplankton functional types exhibit diverse responses to thermal change. *Nature Communications* 12. Springer US: 6413. doi:10.1038/s41467-021-26651-8.

Boyce, D. G., M. R. Lewis, and B. Worm. 2010. Global phytoplankton decline over the past century. *Nature* 466. Nature Publishing Group: 591–596. doi:10.1038/nature09268.

Eppley, R. W. 1972. Temperature and phytoplankton growth in the sea. *Fishery Bulletin* 70: 1063–1085.

---

## Author Comment (AC1)

**Asselot et al. – A missing link in the carbon cycle: phytoplankton light absorption.**

**Summary**

Phytoplankton light absorption is the process by which incident short wave radiation is absorbed by phytoplankton in the ocean's surface waters. This process increases the amount of heat that is absorbed in surface layers while decreasing the depth to which this radiation penetrates. Because phytoplankton also decrease the albedo of the ocean, the net effect is that more shortwave radiation is absorbed in the ocean, specifically the near-surface waters, when more phytoplankton are present.

Asselot et al. present experiments that explore the effect of phytoplankton light absorption on key climate variables of societal importance: ocean surface temperatures, phytoplankton biomass, atmospheric CO2 concentrations, and surface atmospheric temperatures. They do so on the global scale. To do so, they use an Earth System Model of Intermediate Complexity (an ESMIC) called the EcoGEnIE. The ocean-biogeochemical-ecosystem components of the model are equipped with phytoplankton light absorption, such that experiments can be performed with this process turned on or off.

Asselot et al. acknowledge that other studies have already undertaken modelling experiments that assess the effect of phytoplankton light absorption under climate change scenarios. The novelty of their study is therefore in (1) its use of multiple, extended RCP scenarios (2.6, 4.5, 6.0 and 8.5), and (2) its insights into the effect on atmospheric CO2 concentrations since they use emissions, rather than specifying the atmospheric CO2 concentration.

The main findings of this study are:
• That phytoplankton light absorption increases surface temperatures and phytoplankton biomass.
• That the effects are strongest under the weaker global warming scenarios.
• That the warming of the surface ocean weakens the ocean's ability to absorb CO2, which increases atmospheric CO2 and thereby increases surface temperatures in a positive feedback response.

We would like to thank the referee for the very thoughtful and constructive comments.

**Major comments**

This paper is presented simply, and in that sense the authors are honest with their results. They do not oversell the lessons. However, this suggests that the mechanisms involved are very simple, and actually I am left wanting more. I thus have three major concerns with this paper as it stands. All relate to the discussion of their results.

The three main concerns of the referee will be addressed in the following paragraphs.

First, the authors do not explain the mechanisms (physical and/or biogeochemical) that underly the relatively greater chlorophyll biomass when phytoplankton light absorption is included. As far as I can tell from the work, they speculate about the mechanism, but do not definitively show it. If upwelling

increases, why? I think the paper would be strongly improved with a more concrete explanation of this process. I have suggested that the authors undertake some 1D modelling work in my specific comments below, but if they can identify the mechanism with output from the 3D model than that is also fine. Bottom line, the reader needs a clearer more convincing explanation.

The higher surface chlorophyll biomass with phytoplankton light absorption is due to two different mechanisms. First, the dynamics associated with phytoplankton light absorption leads to a weaker biological pump, leading to more labile inorganic matter (e.g. DIC) at the surface of the ocean (Asselot et al., 2021, JAMES). As a consequence, the remineralization is enhanced and the nutrient concentrations increase at the surface. Second, phytoplankton light absorption increases the oceanic temperature along the whole water column, leading to more energy being stored in the ocean. As a consequence, upward vertical velocity is enhanced in the upwelling and mid-latitudes regions. This physical process brings more nutrients at the ocean surface. All in all, these two mechanisms explain the higher surface chlorophyll concentration.
Two sentences are added in the "surface chlorophyll biomass" section.

Second, the real novelty of this study is in the feedback on CO2 concentration, as the authors state in the introduction. No other modelling study (to theirs and my knowledge) has specifically targeted this feedback. However, the consequences for atmospheric CO2 are reported but the mechanisms are not explained. Why exactly does phytoplankton light absorption increase atmospheric CO2? What I'm really asking here is what portion of the atmospheric CO2 increase can be apportioned to a decrease in the solubility pump, a decrease in the biological pump or a decrease in the carbonate pump (assuming this is included)? I have little doubt that the increase in surface temperature is the main culprit, but quantifying these terms would significantly improve the paper.

In a previous study (Asselot et al., 2022); we estimate that the decrease in CO2-solubility due to warmer SST via phytoplankton light absorption enhances the air-sea CO2 fluxes by roughly 10%. The changes in the other mechanisms such as the biological pump and the carbonate pump only increase the air-sea CO2 fluxes by <1%. We find by far that the solubility pump has the largest effect on the increase of atmospheric CO2 concentrations.
A sentence is added to the "synthesis" section.

Third, the discussion is very short. There is no discussion of which models include this feedback as part of their architecture and which do not. There is little discussion of observational studies that see this in the real ocean, nor a discussion of whether the mechanisms that cause a relative increase in chlorophyll biomass are realistic. There is little discussion regarding how phytoplankton community composition changes might affect the magnitude of phytoplankton light absorption in the future (i.e. cyanobacteria becoming dominant). There is also no acknowledgement of the uncertainty in future primary production, which may increase, remain stable or decrease depending on the model and region of interest. Would phytoplankton light absorption cause substantially greater greenhouse warming if global warming was coincident with stable or increasing global primary production? To what degree would the solubility pump outweigh the oceanic gains in carbon from the biological pump? These sorts of discussion points would be highly beneficial to this paper.

We add a complete new paragraph in the "discussion" section of the revised manuscript to answer the concerns of the reviewer.

Finally, the appendices are not really appendices. They are single sentence additions, and they really shouldn't be appendices if they can easily be added to the main text.

We modified the appendices by removing the Appendix 4 that could easily be added to the main text.

**Specific comments**

• Line 21: "decreases or will decrease" isn't that the same thing?

We removed "will decrease".

• Line 23: Using (Boyce et al. 2010) reference is not recommended. See the responses to this paper in Nature underlining its caveats (https://www.nature.com/articles/nature09951)

We remove the sentence with Boyce et al., 2010 and rather introduce the study of Boyce et al., 2014.

• Line 27: -2.99 ± 9.11 is not significantly different from zero. The overall response of the oceanic primary production to global warming is highly uncertain. I would stress that as biogeochemical models have evolved from CMIP5 to CMIP6, the response of phytoplankton biomass and Net Primary Production are more uncertain. This doesn't undermine your study, but it I think it is important to tone down the confidence with which you are projecting a decline.

We add the sentence "This estimate is not significantly different from zero due to the evolution of biogeochemical models from CMIP5 to CMIP6, thus the response of phytoplankton biomass is more uncertain."

• Line 43: "… reports a decline of chlorophyll concentrations associated with a local oceanic warming of up to 0.7 °C. This maximum warming is attributed to changes in ocean circulation, under the global warming scenario". This could be written more clearly. I am unsure of what you mean.

We re-phrase the sentence.

• Line 75: typo in model

Changed

• Line 144: It is unclear what is done following the spin-up. What years is the 737-year run and which years it is running from? If the run begins at 1765, then your total number of years is 736, not 737.

First, we run a 10,000 years spin-up with BIOGEM only. The spin-up is used as a "restart file" for the 8 simulations, thus the simulations have a realistic nutrient distributions when they start. Second,

following the spin-up, we run our simulations with ECOGEM. The runs begin at 1765 so the total number of years is indeed 736 years. All the simulations consider ECOGEM.

• Line 150: It is odd to consider your model validation as being a comparison with other models. I would advocate that you either compare your model with observations, or your change the title of this section to "model inter-comparison".

We changed the title of the section.

• Line 151: EMIC has not yet been defined.

EMIC is previously defined in the "model description" section

Section 4.1.1: Here I am a little dissatisfied with the explanation of why chlorophyll concentrations are greater in the simulation with phytoplankton light absorption. All else being equal, if more shortwave radiation is absorbed in the upper layers of the ocean and less is able to penetrate down to deeper layers, then this should increase the vertical density gradient through the upper water column. An increase in stratification is the result, limiting vertical mixing of nutrients to near-surface layers. However, I suppose that if less radiation penetrated to deeper layers, then the strongest density gradients would exist nearer the surface, lessening density gradients deeper in the water column and thereby increasing the nutrient flux to the lower euphotic zone? All in all, I think I need a clearer explanation of the physical mechanism that is occurring. It would be informative and highly beneficial to this paper to conduct 1D water column simulations that test the effect of phytoplankton light absorption on the nutrient fluxes to phytoplankton.

Although a 1D model is easier and may give concrete answers, several previous model studies (e.g. Asselot et al., 2021, Paulsen et al., 2018) show that in an ESM with an atmospheric component the dynamics are different. In our previous article we have shown that the larger surface chlorophyll concentration associated with phytoplankton light absorption is due to two different mechanisms. First, phytoplankton light absorption leads to weaker biological pump and a larger amount of labile inorganic matter at the surface, enhancing the remineralization at the surface. As a consequence, the higher remineralization leads to larger nutrient concentrations at the surface and thus enhances the surface chlorophyll biomass. Second, the upward vertical velocity, specifically in the upwelling and mid-latitude regions, is enhanced due to the global warming of the ocean. As a consequence, the penetration depth of sinking material is reduced and organic matter is trapped closer to the surface. The combination of these biogeochemical and physical mechanisms explains the higher surface chlorophyll biomass with phytoplankton light absorption.

• Line 205: "not been tuned yet". This suggests that the model has never been tuned. Surely that's not correct.

The model has been previously tuned to get reasonable primary production and nutrient fields but not to match the projected atmospheric CO2 concentrations.
We rephrase our sentence.

• Section 4.2.1: What component of the increase in atmospheric CO2 is due to (i) the solubility pump, (ii) the carbonate pump and (iii) the biological pump? Just saying that the increase in CO2 is due to additional surface heating is not convincing. I am not saying that the majority of it is due to biological mechanisms, but surely a portion of the change is due to the changes in phytoplankton biomass being closer to the surface ocean, and therefore not exporting organic carbon to deeper levels as efficiently? Alternatively, the increase in phytoplankton biomass may in fact tend to increase the air-sea flux of CO2, but this is more than opposed by the solubility effect of additional warming.

With our model setup, we already showed that the increase in atmospheric CO2 concentration is mainly due to the solubility pump (Asselot et al., 2022). The changes in solubility pump enhance the air-sea CO2 fluxes by 10% while the changes in biogeochemical pumps enhance the air-sea CO2 fluxes by <1%. Clearly, the solubility pump has the largest effect on the increase atmospheric CO2 concentration with phytoplankton light absorption. Furthermore, the warmer ocean leads to a reduced ocean CO2 uptake, explaining in part the increase in atmospheric CO2 concentration with phytoplankton light absorption (see response to referee #3).
We add sentences in the "synthesis" section.

• Line 233-236: These mechanisms have been proposed to explain to explain the increase in phytoplankton biomass in the runs with phytoplankton light absorption, but they have not been explained mechanistically or shown.

We explain more in detail the mechanisms behind the increase in surface chlorophyll concentration in the "oceanic properties" section.

• Line 249: The limitation of phytoplankton growth at temperatures greater than 20 °C is an odd choice… What is the motivation for this choice in parameterisation? Because this parameterisation is really quite arbitrary and probably not realistic (i.e. does not follow the Eppley curve (Eppley 1972) nor the recent work by Anderson et al. (Anderson et al. 2021) that confirms community-wide exponential increases in growth rates with temperature), I have to encourage the authors to discuss how this result for RCP8.5 simulations is likely not realistic either. Also, please tell us what the parameterisation is. What is the equation?

The equation of the temperature limitation is an Arrhenius-like equation:
$$\gamma_T = e^{A(T - T_{ref})}$$
With $\gamma_T$ is the temperature limitation, A is the temperature sensitivity, T is the sea surface temperature and $T_{ref}$ is the reference temperature.
In the model setup, $T_{ref}$ = 20°C thus if the SST exceeds 20°C then the temperature limitation increases and phytoplankton growth is limited. $T_{ref}$ is set to 20°C because most experimentally determined rates are done at 20°C. Additionally, several experiments with different phytoplankton communities indicate that the maximum growth rate is reached at 20°C and exceeding this value limits phytoplankton growth (e.g. Goldman, 1977; Rhee and Gotham, 1981). Therefore we consider that $T_{ref}$ = 20°C is realistic.
We add sentences in the "ecosystem component" section.
We also discuss that there may be changes if adaptation was implemented in the model setup.

• Line 250: Additionally, the limitation is definitely present in all your scenarios, but RCP8.5 is likely the only one where enough surface grid cells exceed 20 °C that it has a significantly negative effect on the chlorophyll/temperature increase. It is worth mentioning that this is the case. In fact, it may explain the slight decreases in the phytoplankton light absorption effect for each RCP scenario from 2.6 to 4.5 to 6.0.

Indeed the limitation is present in all the simulations but only under the RCP8.5 scenario there are enough grid cells exceeding 20°C. This explanation is added to the revised manuscript.

• Paragraph beginning Line 276: This paragraph discusses the stronger effect of phytoplankton light absorption in RCP scenarios 2.6, 4.5 and 6.0, with RCP8.5 showing a weaker role. The authors take this opportunity to explain that this effect is weaker in RCP8.5 because stronger warming causes an increase clarity (by reducing phytoplankton biomass) and therefore the phytoplankton absorption effect is weaker. But, the authors have only just discussed that the weaker effect in RCP8.5 is due to the arbitrary limitation of phytoplankton above 20 °C. The different response in RCP8.5 is therefore an artefact of the model, and is not realistic. Therefore, it is not evidenced by your study to say that "our findings indicate that a severely warmer world increases ocean clarity and slows down the phytoplankton-induced global warming".

As argued previously, the temperature limitation of phytoplankton growth above 20°C is realistic. The difference response under the RCP8.5 scenario is thus realistic. As a consequence the main conclusion of this study remains identical.

Thank you for considering my input to your research.

Pearse J. Buchanan

**References**

Asselot, R., Lunkeit, F., Holden, P. B., & Hense, I. (2021). The relative importance of phytoplankton light absorption and ecosystem complexity in an Earth system model. Journal of Advances in Modeling Earth Systems, 13(5), e2020MS002110.

Asselot, R., Lunkeit, F., Holden, P. B., & Hense, I. (2022). Climate pathways behind phytoplankton-induced atmospheric warming. Biogeosciences, 19(1), 223-239.

Boyce, D. G., Dowd, M., Lewis, M. R., & Worm, B. (2014). Estimating global chlorophyll changes over the past century. Progress in Oceanography, 122, 163-173.

Goldman, J. C. (1977). Temperature effects on phytoplankton growth in continuous culture. Limnology and Oceanography, 22(5), 932-936.

Paulsen, H., Ilyina, T., Jungclaus, J. H., Six, K. D., & Stemmler, I. (2018). Light absorption by marine cyanobacteria affects tropical climate mean state and variability. Earth System Dynamics, 9(4), 1283-1300.

Rhee, G. Y., & Gotham, I. J. (1981). The effect of environmental factors on phytoplankton growth: temperature and the interactions of temperature with nutrient limitation. Limnology and Oceanography, 26(4), 635-648.

Anderson, S. I., A. D. Barton, S. Clayton, S. Dutkiewicz, and T. A. Rynearson. 2021. Marine phytoplankton functional types exhibit diverse responses to thermal change. Nature Communications 12. Springer US: 6413. doi:10.1038/s41467-021-26651-8.

Boyce, D. G., M. R. Lewis, and B. Worm. 2010. Global phytoplankton decline over the past century. Nature 466. Nature Publishing Group: 591–596. doi:10.1038/nature09268.

Eppley, R. W. 1972. Temperature and phytoplankton growth in the sea. Fishery Bulletin 70: 1063–1085.

---

## Author Comment (AC2)

**Comment on esd-2021-91**

Anonymous Referee #2

Referee comment on "A missing link in the carbon cycle: phytoplankton light absorption"
by Rémy Asselot et al., Earth Syst. Dynam. Discuss.,
https://doi.org/10.5194/esd-2021-91-RC2, 2022

Phytoplankton in the ocean absorbs light, and the amount of phytoplankton exists changes the degree of shortwave penetration into the ocean. Since shortwave radiation raises water temperature, there may be differences in water temperature, primary production, and climate change if phytoplankton light absorption is taken into account in water temperature fluctuations in a model or not. Using an Earth system model of intermediate complexity, the authors conducted future scenario experiments with and without phytoplankton light absorption to evaluate how phytoplankton light absorption would affect primary production and climate change. I believe that this study points out what has been lacking in conventional climate research and provides important implications for future model development. However, before accepting this manuscript, I think that there are some aspects of the authors' analysis that could be improved as follows. I may have misread some things, and I do not think the authors need to follow all my comments, but I hope that the following comments will help the authors to improve their manuscript.

We would like to thank the referee for the constructive suggestions.

1.
Model validity
l.154 "Figure 3 shows that our increase in SAT are in agreement with Zickfeld et al. (2013)"
In order to show the validity of the model, the authors compared the change in SAT with a previous study, but what about the distribution of SST, nutrients, chlorophyll concentration, etc.? Since these are directly relevant to this study, I think it would be better to compare climatic values, etc. with observations to see if the model reproduces the approximate distribution.

In a previous study (Asselot et al., 2021, JAMES) we compare our modelled AMOC, surface chlorophyll concentration, primary production, export production and phosphate concentration with observations. We show that our model reproduces suitable patterns and distributions for these climate variables.

Mechanism enhancing upwelling
l.180 "an enhanced vertical velocity which is triggered by changes in the oceanic heat budget"
In this model, the shortwave penetrates down to the sixth layer at 221.84 m, but the authors insisted the upwelling at 326 m depth is enhanced due to the phytoplankton light absorption (l.180). It is not clear why the upwelling is enhanced.

In our model setup, phytoplankton light absorption warms the whole water column. As a consequence, more energy is stored in the ocean, increasing the oceanic velocity and thus the vertical velocity in the upwelling regions. This explanation is added to the revised manuscript.

l.235 "this biogeophysical mechanisms enhances the upward vertical velocity"
I could not understand why that would happen.

Phytoplankton light absorption increases the oceanic temperature along the whole water column. As a consequence, more energy is stored in the ocean, enhancing the upward vertical velocity.

In addition, the concentration of chlorophyll depends not only on nutrients but also on water temperature and shortwave radiation. It would be good to have a discussion of how these factors might have affected the results.

Indeed, the concentration of chlorophyll depends on nutrients, temperature and shortwave radiation. However, phytoplankton light absorption doesn't affect incoming shortwave radiation thus this factor doesn't affect our results. Temperature affects the chlorophyll concentration but, on a global scale, it only affects phytoplankton growth for the simulations following the RCP8.5 scenario. Similar discussion is added to the "synthesis" paragraph.

Difference from the concentration runs
l.213 "However, in Asselot et al. (2021a) we do not prescribe any CO2 emissions, neglecting their effect on the atmospheric CO2 concentration."
It is ambiguous why the increase in atmospheric CO2 concentration of the emission driven runs is different from that of the concentration-driven runs, so it would be better to add some analysis. For example, by considering the phytoplankton light absorption, can we estimate how the water temperature changes, how it changes the carbon concentration in the ocean and the atmosphere-ocean carbon flux, and how it changes the concentration of the atmospheric CO2? If the authors could figure this out, it would clarify the difference from concentration driven experiments and the importance of this study.

With this sentence, we compare the results of Asselot et al. (2021) where CO2-emissions are not prescribed versus our actual results where CO2-emissions are prescribed. In both studies we analyse how phytoplankton light absorption affects the water temperature and its consequences on the carbon cycle. In our actual study, the changes in chlorophyll biomass, SST, SAT and atmospheric CO2 concentrations are higher than in Asselot et al. (2021), except under the RCP8.5 scenario, which is a particular case. The only difference between the two studies is the prescribed CO2-emissions in the actual study, explaining the higher changes in atmospheric CO2 concentration in our actual study.

Results and analysis on RCP8.5 runs
l.249–251 "This is due to the model setup where a SST higher than 20C limits phytoplankton growth. This threshold is only exceeded for the simulations RCP8.5-LA and RCP8.5 (Appendix D1), therefore phytoplankton growth is limited by the ocean temperature in these two simulations (Figure 10)."
First, I think that this 20 degreeC limit is arbitrary (part of tuning), so I am not confident that the results of the RCP8.5 experiments relying on it are correct. Even in the current climate, water temperatures are above 20 degreeC in equatorial regions, etc. Is this high SST in these regions not reproduced in the model? I do not believe this is the case, so the authors may want to reconsider their analysis for RCP8.5 runs.

The limitation of phytoplankton growth is set to 20°C because most experimentally determined rates are done at 20°C. Additionally, several experiments with different phytoplankton communities indicate that the maximum growth rate is reached at 20°C and exceeding this value limits phytoplankton growth (e.g. Goldman, 1977; Rhee and Gotham, 1981). However, in a warmer world, adaptation may happen and this point is added in the discussion.

The high SST in tropical regions is reproduced and the 20°C threshold is exceeded in all the simulations. But only in the simulations *RCP8.5-LA* and *RCP8.5* enough surface grid cells exceed 20°C to affect the global increase in chlorophyll biomass.

We rephrase our sentence.

Figure 10

How was this calculated? Was it calculated from the global-mean SST? If so (see Table D1), I do not understand why the temperature limitation is calculated from the global mean SST, since the growth rate of phytoplankton is determined by the water temperature and other conditions at each location.

Indeed, the values of Figure 10 were calculated from the global-mean SST. We took the global-mean SST to calculate the mean temperature limitation around the global ocean. We focus on global estimates rather than local estimates due to the limitations of EcoGEnIE which is a model of intermediate complexity.

As mentioned above, there are some points that need to be improved regarding the analysis of the results. Hopefully the authors will revise the manuscript to make it more convincing.

Some minor comments are listed below. I hope they will be useful for the authors to revise their manuscript.

Minor comments:

Section 2.1

There is no description of the emissions from 2100 to 2250, so it would be good to describe them briefly.

We add a sentence to describe the emissions between 2100 and 2250.

l.75 "The Earth system mode (ESM)l"

This should be "The Earth system model (ESM)"

Changed.

l.78–82

I think the authors want to show how much EcoGEnIE was used, and for that, it would be better to use past tense or present perfect tense.

We re-phrase our sentences.

Section 2.2.4
It would be easier to understand if the previous method without phytoplankton light absorption is also described and compared.

In the previous version of the model without phytoplankton light absorption, light was only absorbed by phytoplankton. In our model version, a new light scheme is implemented where the absorbed light by phytoplankton is converted into heat and is able to affect the oceanic temperature.
Implementing the new light scheme with phytoplankton light absorption increases the surface chlorophyll concentration, the SST, the atmospheric $CO_2$ concentrations and the atmospheric temperature. For a more detailed comparison of the two model version, we refer to Asselot et al. (2021, JAMES).

l.136 "dT/dt denotes the temperature changes"
The dT/dt term here is the water temperature change only associated with radiative heating, so I think it should be clearly stated as such.

Changed.

l.139 "(Asselot et al., 2021a)"
This should be "Asselot et al. (2021a)"

Changed.

l.143 "The spin-up is run with a constant pre-industrial atmospheric $CO_2$ concentration of 278 ppm."
l.154 "In the pre-industrial simulation, the atmospheric $CO_2$ concentration is constant and set to 278 ppm."
I thought the model was emission driven, but do these sentences mean that it is concentration driven during spin-up? Instead of a spin-up with zero emissions? If so, when is the timing of changing from concentration driven to emission driven? Was there any shock at the timing when it became emission driven?

The atmospheric $CO_2$ concentration is only prescribed for the spin-up phase. The spin-up in run for 10,000 years and our simulations are run for 736 years (sorry, we accidentally wrote 737 years) with emission driven atmospheric $CO_2$ concentration. Looking at our outputs, there was no shock when we changed from concentration-driven to emission-driven atmospheric $CO_2$ concentrations.

l.144 "after the spin-up for 737 years-long, since the $CO_2$ emissions data are only available for this time span"
Does this 737 years refer to the past 737 years? Or does it include future data? I think it is unclear how the spin-up was done after ECOGEM was switched on, so it would be good if the authors could write it clearly.

The 736 years-long simulations represent past and future data, from the year 1765 to the year 2500. This information is added to the manuscript.

l.147 "In total, we run 8 similations following the RCP scenarios"
Before the RCP scenarios, there should be a historical run (Figure 1), but the description of the method for the historical run is ambiguous.

We re-phrase the sentence

l.176 "This pronounced chlorophyll increase is due to the coarse grid resolution"
Why does chlorophyll increase if the grid resolution is coarse?

The dimension of the grid doesn't affect the chlorophyll concentration. Rather, the sharp patterns of the chlorophyll concentrations are explained by the coarse grid resolution.
We revise our sentence.

l.178–179 "the upwelling and mid-latitude regions" / "the subtropical gyres"
The upwelling, mid-latitude, and subtropical regions can be overlapping, so the meaning of the sentence is unclear.

We revise our sentence.

l.186 "the SST is the zonally-averaged temperature from the surface to 29 m depth"
"vertically-averaged"?

Changed.

l.204 "the expected atmospheric CO2 concentrations in 2500 under the RCP scenarios are
not reached."
It is difficult to understand for me, I am glad if the authors rewrite it.

Changed.

l.205 "EcoGEnIE has not been tuned yet."
Are the authors saying that more tuning is needed? Since the concentrations in emission driven runs are the result of various balances and it is difficult to adjust the concentrations in the model to reality, why not simply state the fact that atmospheric CO2 concentrations are low here?

Yes we suggest that our version of the model with primary production allowed until the sixth layer of the model should be tuned to match, at least the pre-industrial CO2 level in all the simulations.

l.217 "The global changes in oceanic and atmospheric properties due to phytoplankton light absorption lead to an increase of surface atmospheric temperature (SAT) (Figure 8)."
I believe that the importance of plankton light absorption can be conveyed to readers by clearly describing the mechanism of how plankton light absorption affects SAT through the changes in ocean and atmosphere conditions.

We add a sentence to explain the main mechanism changing the atmospheric temperature.

l.234 "increasing the remineralization and thus the nutrient concentrations at the ocean surface"
Since there has been no mention of remineralization before, it is difficult to follow the discussion.

In the revised version of the manuscript, the remineralization process is mentioned earlier.

l.241 "the sea-air CO2 flux"
This is not clear whether the authors are referring to upward flux or ocean carbon uptake, so please specify.

Changed.

l.259 "the model has not been tuned in this configuration yet."
See comments on l.205.

We suggest tuning our version of the model to have match at least pre-industrial levels.

l.286 "phytoplankton-induced global warming"
I do not think phytoplankton will induce global warming. I can see how the light absorption of phytoplankton can affect the progress of global warming, and in my opinion, the word "induced" is not appropriate here.

We rephrase our sentence

Figure 3
Does this figure represent global mean SAT changes?

It represents the global mean SAT changes between the periods 1986-2005 and 2281-2300.

Figure 4 caption "Zonally"
"Globally"?

Changed

**Reference**

Asselot, R., Lunkeit, F., Holden, P. B., & Hense, I. (2021). The relative importance of phytoplankton light absorption and ecosystem complexity in an Earth system model. Journal of Advances in Modeling Earth Systems, 13(5), e2020MS002110.

Goldman, J. C. (1977). Temperature effects on phytoplankton growth in continuous culture. Limnology and Oceanography, 22(5), 932-936.

Rhee, G. Y., & Gotham, I. J. (1981). The effect of environmental factors on phytoplankton growth: temperature and the interactions of temperature with nutrient limitation. Limnology and Oceanography, 26(4), 635-648.

---

## Author Comment (AC3)

This paper describes a set of long (> 500 y) simulations with an EMIC, with and without phytoplankton absorption of solar radiation and the associated ocean heating, and argues that this represents an important and mostly neglected process in the climate system. What is novel and interesting is that (1) phytoplankton biomass tends to increase rather than decrease under enhanced greenhouse forcing, (2) the effect is not monotonic with increasing emissions, and tends to be damped or reversed in the highest emission scenario examined, and (3) the effect on atmospheric CO2 concentration appears to be large.

I believe that this is an important experiment that deserves to be published, but the paper is not well written and requires, at least, major revision. Possibly it would be better if the editors declined the paper and returned it to the authors so that they could take the time required to craft a more substantial contribution. The English is adequate, but it would be best if the authors could find a native-speaker colleague to give it a thorough English editing before resubmission. The title could be revised to be more specific about what the actual content of the paper is; the present wording is fairly generic and uninformative.

We would to thank the reviewer for his constructive and instructive comments.

Major points:

(1) The authors do not make a lot of effort to explain the mechanisms underlying the differences observed between the Light Absorption (hereafter LA) case and the non-LA case. Combined with the inadequacies of the model description, this makes it difficult to credit some of the more dramatic claims made, in particular the much higher atmospheric CO2 concentration in the LA case.
I estimate that by temperature-dependence of CO2 solubility alone, an increase in the global mean ocean temperature of 1 C would increase atmosphere CO2 by ~14 PgC. In the experiments shown here the increase in SST (which would be an extreme upper limit for the change in the ocean mean) is <1 C (Figs 4+6, Table D1). A weaker biological pump could add another ~5 PgC, assuming Delta-DIC/Delta-PO4 = 106 and a net increase of ~0.1 mmol P m^-3 (Table C1) over a surface layer 100 m thick. The atmospheric CO2 increase shown here is extremely large by comparison, ranging from ~75 PgC in RCP2.6 to ~250 Pg in RCP8.5, assuming that it takes ~2 PgC as CO2 to increase the atmosphere concentration by 1 ppm (Figure 7).
Maybe these are simplistic, static calculations. The experiments are long and the ocean is continuously overturning, so maybe the explanation lies in upwelling of the existing ocean inventory of DIC into a surface ocean that is getting warmer and therefore outgassing more CO2 to the atmosphere. A problem with this hypothesis is that RCP8.5 has by far the largest amount of excess atmospheric CO2 associated with LA (Figure 7), but the smallest increase in SST (Figure 4, Table D4). However, RCP8.5 also has the largest cumulative CO2 uptake (both LA and non-LA), so maybe this could be explained by greater outgassing of anthropogenic CO2 taken up earlier. Possibly the authors could consider comparing the 3D distribution of DIC at the beginning and the end of the experiments, or making a map of net outgassing of CO2. At any rate, I think more effort to identify the underlying mechanisms is warranted. It could also help if they cited some previous publications that demonstrate that this very coarse-resolution model can produce at least approximately realistic ocean upwelling.

We agree and don't think that the temperature dependence of solubility alone cannot explain changes of up to 250 PgC in steady state. Furthermore, we showed in a previous study that enhanced upwelling only slightly affect the atmospheric CO2 concentration with phytoplankton light absorption (Asselot et al., 2021).
To answer the concern of the reviewer, we compute the difference of atmospheric CO2 concentrations between the PLA case and the non-PLA case and plot them against the CO2 baseline (i.e. values of the non-PLA case).

[Figure]

As shown on the figure above, for the first three RCPs scenarios (but not RCP8.5), the change in atmospheric CO2 concentration driven by phytoplankton light absorption follows a roughly linear dependence on the baseline concentration for that RCP. This suggests a transient effect which is driven by reduced CO2 uptake rates that are limited by CO2 solubility. The rate of CO2 uptake is roughly proportional to baseline concentration for the first three RCPs scenarios but is reduced for RCP8.5 because of the smaller phytoplankton light absorption effect on SST. To validate this inference, we continue our simulations for another 1000 years with no further CO2 emissions.

[Figure]

As shown on the second figure, we clearly see that the CO2 differences decrease through time, converging towards the far smaller steady-state difference previously highlighted by Asselot et al. (2021). This result evidences that large CO2 differences are driven by a transient effect of reduced CO2 uptake fluxes, consistent with reduced CO2 solubility under phytoplankton light absorption warming.
These figures and explanations are added to the revised manuscript.
We also cite previous publication to justify the rather realistic upwelling.

(2) There are some important details missing from the model description. I understand that all of the submodels are previously published, but key details that are directly relevant to the results presented should be briefly reiterated. Most disturbingly, the assertion that the ocean biology model operates in a static fashion at each grid point, without ocean transport of the related tracers, appears from nowhere in the Results (198), whereas the Methods appears to say the opposite (90). One could imagine, for example, that increased upwelling of nutrients causes large increases in phytoplankton biomass at the grid points where upwelling occurs (with the additional heat being advected away), in a way that might not occur if the phytoplankton were also being advected by

ocean currents. Could they show some profiles of how chlorophyll concentration evolves over time? Do they remain within the range of historical observed values? Or at least within the realm of plausibility? The paper also appears not to state whether chlorophyll concentration in the non-LA expts is given a constant, nonzero value, or is assumed to be 0 (k=k_w).

We add a sentence in the revised manuscript to explain that the state variables of the ecosystem component are not transported. As pointed out by the reviewer, we also add a sentence in the "chlorophyll biomass" section to discuss what would happen if phytoplankton could be advected.
In our previous study (Asselot et al., 2021), we compared the observed and modelled surface chlorophyll concentration (see maps below). The global pattern of surface chlorophyll biomass is in agreement with the satellite-derived estimates. The high latitudes show a large chlorophyll biomass while the subtropical gyres indicate a low chlorophyll biomass. However, the model underestimates the magnitude of the surface chlorophyll biomass. This is particularly true in the northern polar region and the upwelling regions. These limited agreements with observations are in line with the results of Ward et al. (2018).

[Figure]

In the simulations without phytoplankton light absorption the chlorophyll concentration is free to evolve. However, in these simulations, $k_{chl} = 0$ which mean that $k = k_w$

There is no description of parameterizations of phytoplankton photoacclimation or chlorophyll synthesis or degradation. As I understand it, the model has prognostic phytoplankton C, P, and chlorophyll (117). So there is no photoacclimation per se: both C and Chl are prognostic. But a brief description of the phytoplankton growth and chlorophyll synthesis model is warranted, and a statement of whether there is any loss of chlorophyll independent of grazing or other loss of cells. Chlorophyll synthesis requires N and Fe, but not P. I assume they use a fixed N/P ratio to estimate the dependence of chlorophyll synthesis on N; whether it also depends on Fe is not stated. Nor is it stated whether the C/P and C/Fe ratios are fixed or variable; I assume that C/Fe is fixed and C/P variable, as there is prognostic phytoplankton P but not Fe. This should be clearly explained and ratios used (where they are fixed) stated.

The model includes a dynamic photoacclimation following Geider et al. (1998), the Chl:C ratio can vary depending on light availability. The model considers nutrients (DIC, PO4 and Fe), plankton biomass and organic matter (POM and DOM) as state variables. Phytoplankton growth is limited by light, temperature and nutrient availability. The model assumes that photosynthesis is a Poisson function of irradiance and that phytoplankton growth is limited though this function (Geider et al., 1998; Moore et al., 2001). Nutrient uptake is a Michaelis-Menten function of nutrient concentration and phytoplankton growth is limited by a minimum function of internal nutrient status. Temperature limits phytoplankton growth through an Arrhenius relation by affecting light-saturated photosynthesis and maximum nutrient-uptake rates. Phytoplankton biomass is only lost via grazing and mortality. Indeed the model considers a fixes N:P ratio of 16 for photosynthesis. However the

C:Fe and C:P are flexible meaning that phytoplankton can flexibly take up nutrients according to availability.

Nor is there any description of surface solar irradiance or of radiative transfer in the atmosphere. It is stated that the incoming shortwave varies seasonally (140), but there is nothing about its geographic distribution (for example, top-of-atmosphere irradiance might be calculated from astronomical formulae and atmospheric attenuation assumed constant). They should also specify the fraction of total solar irradiance that is assumed to be shortwave/longwave, as only the former is affected by phytoplankton absorption. The energy balance atmosphere presumably has a submodel for radiative transfer (e.g., how does upwelling/downwelling longwave radiation vary as a function of atmospheric $CO_2$ concentration). The climate changes are strongly dependent on this, so at least a brief description is warranted.

The incoming shortwave radiation at the top of the atmosphere is calculated from astronomical formulae including the planetary albedo. The planetary albedo varies as a function of latitude and time of year to account for the effects of changes in solar zenith angle. The atmospheric attenuation is indeed constant. The net longwave radiation represents 45% of the total atmospheric energy balance while net shortwave radiation represents 25%. The outgoing planetary longwave radiation is parameterized to implement the radiative forcing associated with changes in atmospheric $CO_2$ concentrations. Higher atmospheric $CO_2$ concentration leads to higher amount of outgoing shortwave radiation being trapped in the atmosphere.
These explanations are added in the revised manuscript.

Other less immediately relevant process that could use a brief description include carbon chemistry and gas exchange (106-107), the wind data used and the calculation of the wind stress and the drag coefficient (the non-dynamical energy balance atmosphere requires that wind speed and wind stress at the ocean surface be specified), and the dependence of ocean vertical mixing on stratification.

The air-sea gas exchange depends on the gas transfer velocity, the water density, the concentration of dissolved gas in the surface ocean, the solubility coefficient calculated from Wanninkhof (1992), the concentration of gas in the atmosphere, and the fraction of the ocean covered by sea ice.
Wind is considered as an external factor in the model and is prescribed for all simulations. The model uses the annual average wind velocities of Trenberth et al. (1989). The prescribed wind stress is the monthly wind stress climatology of Kalnay et al. (1996) reanalysis data. The drag coefficient is set to $1 \times 10^{-3}$ (Weaver et al., 2001).
Finally, an enhanced stratification in the ocean leads to a reduced vertical mixing.
We add these explanations in the revised manuscript.

Finally, the description of the spinup and the experimental design is confusing. First they spun up the model for 10000 years with BIOGEM but not ECOGEM "to have a realistic distribution of nutrients" (142-143). Then there is possibly a further spinup with ECOGEM turned on, before the historical/RCP experiments are launched, but the description is confusing and I can't really tell what was done. Why would spinning up the model produce a realistic distribution of nutrients if there is no biological pump? I would have spun it up for a further 2000-3000 years with all of the biological processes active. Nor is it stated how they know that the system is in steady-state at 2500 (149).

First we run the model for 10,000 years with BIOGEM only. During this spin-up phase the realistic distribution of nutrients is achieved because BIOGEM considers an implicit biological pump. This component doesn't explicitly resolve the biological community and instead transforms surface inorganic nutrients directly into exported nutrients or dissolved organic matter. Second, ECOGEM is switched on and the simulations are launch for 736 years (from 1765 and 2005).

We apologize for the inconsistency but after double-checking; the climate system is not in steady-state. The CO2 emissions are prescribed all along the simulations, thus looking at outputs of the year 2500 means that the climate is not in steady-state. However, most of climate projection studies investigate the climate system in a non-steady-state.

(3) The Introduction is a grab-bag of literature citations intended to provide the impression that there is a broad consensus that phytoplankton biomass has declined over the historical period and is likely to decline further in the enhanced-greenhouse future. In my opinion this assertion is nowhere near as robust as the authors imply and gives the main premise of the paper a "straw man" quality. In Kwiatkowski et al., the average decline in NPP is only 3% by 2100, in the highest emissions scenario. Bopp et al (2022; 10.5194/bg-2021-320) suggest that phytoplankton biomass may be a more robust diagnostic than NPP (this paper is still in the Discussion stage, but the authors should at least take a look at it). Boyce et al 2010 drew some rather vigorous criticism (www.nature.com/articles/nature09953). Boyce et al (2014, 10.1016/j.pocean.2014.01.004) address some of these criticisms and should certainly be cited here. They claim that the basic conclusion that a long term secular (downward) trend is detectable remains sound, but this conclusion remains controversial and I think that the authors of the current contribution should treat it a bit more skeptically. The results of Polovina et al and McClain et al represent too short time series to be inferred to represent long-term secular trends, and should be discussed in the context of the difficulty of separating such trends from natural variability (e.g., 10.5194/bg-7-621-2010, 10.1029/2019GB006453). Behrenfeld et al show a statistical relationship between chlorophyll and stratification in the historical record of observed climate variability (mainly ENSO); their extrapolation of this to changes expected under anthropogenic warming is quite speculative (in any case, how can observations (21) tell us what will happen in the future?) I have not read Sonntag or Paulsen, and I can't say I find the synopses offered here very illuminating. As these are PhD theses rather than journal articles it is important to summarize their findings clearly, as the original text may not be accessible to the reader.

We changed the introduction to bring perspective on the results presented in it.
In details, we removed the Boyce et al. (2010) paper and rather introduce Boyce et al. (2014). We also clearly state that the conclusions of Polovina et al. (2008) and McClain et al. (2004) might be altered by their short time series. We rephrase the sentence of Behrenfeld et al. (2006). We slightly rephrase the findings of Sonntag (2013) and Paulsen (2018) but their PhD theses are available online.

Minor points:

Terminology regarding IPCC and the RCPs (57-62): It is a common misconception that CMIPs/RCPs/SSPs are 'commissioned' or 'solicited' or 'approved' by IPCC. Proper citation format for IPCC Assessment (or other) Reports is given in the reports, but citing these in the present context is unnecessary. It is better to just cite Moss et al 2010 (10.1038/nature08823) for the RCPs and Taylor et al 2012 (10.1175/BAMS-D-11-00094.1) for CMIP5. Referring to scenarios as predictions (61) should be avoided (as should referring to scenario-based climate projections as predictions, e.g., 16, 270).

We thank the reviewer for pointing out these misconceptions. We changed the citation format.

I don't think Section 3 is necessary, and it could be folded into the Results. I think this result is worth showing (although it might be better treated as Supplementary). But I think it is overreaching to say that it by itself 'validates' the model setup (and by implication all of the submodels that affect results shown in this paper). The wording should be a bit more tentative and simply describe what was actually tested against what.

We renamed and rephrase section 3

**Reference**

Asselot, R., Lunkeit, F., Holden, P. B., & Hense, I. (2021). The relative importance of phytoplankton light absorption and ecosystem complexity in an Earth system model. Journal of Advances in Modeling Earth Systems, 13(5), e2020MS002110.

Geider, R. J., MacIntyre, H. L., & Kana, T. M. (1998). A dynamic regulatory model of phytoplanktonic acclimation to light, nutrients, and temperature. Limnology and oceanography, 43(4), 679-694.

Kalnay, E., Kanamitsu, M., Kistler, R., Collins, W., Deaven, D., Gandin, L., ... & Joseph, D. (1996). The NCEP/NCAR 40-year reanalysis project. Bulletin of the American meteorological Society, 77(3), 437-472.

Moore, J. K., Doney, S. C., Kleypas, J. A., Glover, D. M., & Fung, I. Y. (2001). An intermediate complexity marine ecosystem model for the global domain. Deep Sea Research Part II: Topical Studies in Oceanography, 49(1-3), 403-462.

Trenberth, K. E. (1989). A global ocean wind stress climatology based on ECMWF analyses. NCAR Tech. note, 93.

Wanninkhof, R. (1992). Relationship between wind speed and gas exchange over the ocean. Journal of Geophysical Research: Oceans, 97(C5), 7373-7382.

Ward, B. A., Wilson, J. D., Death, R. M., Monteiro, F. M., Yool, A., & Ridgwell, A. (2018). EcoGEnIE 1.0: plankton ecology in the cGEnIE Earth system model. Geoscientific Model Development, 11(10), 4241-4267.

Weaver, A. J., Eby, M., Wiebe, E. C., Bitz, C. M., Duffy, P. B., Ewen, T. L., ... & Yoshimori, M. (2001). The UVic Earth System Climate Model: Model description, climatology, and applications to past, present and future climates. Atmosphere-Ocean, 39(4), 361-428.